# Response of Prolyl 4 Hydroxylases, Arabinogalactan Proteins and Homogalacturonans in Four Olive Cultivars under Long-Term Salinity Stress in Relation to Physiological and Morphological Changes

**DOI:** 10.3390/cells12111466

**Published:** 2023-05-24

**Authors:** Aristotelis Azariadis, Filippos Vouligeas, Elige Salame, Mohamed Kouhen, Myrto Rizou, Kostantinos Blazakis, Penelope Sotiriou, Lamia Ezzat, Khansa Mekkaoui, Aline Monzer, Afroditi Krokida, Ioannis-Dimosthenis Adamakis, Faten Dandachi, Boushra Shalha, George Kostelenos, Eleftheria Figgou, Eleni Giannoutsou, Panagiotis Kalaitzis

**Affiliations:** 1Department of Horticultural Genetics & Biotechnology, Mediterranean Agronomic Institute of Chania, Alsyllion Agrokipiou, 73100 Chania, Greece; 2Department of Botany, Faculty of Biology, University of Athens, 15784 Athens, Greece; 3Kostelenos Olive Nurseries, 18020 Poros, Greece

**Keywords:** arabinogalactan proteins (AGPs), AGPs and homogalacturonan immunodetection, Arvanitolia, cell wall, Gaidourelia, Koroneiki, Lefkolia, long-term salinity stress, *Olea europeae*, prolyl 4-hydroxylases (P4Hs)

## Abstract

Olive (*Olea europeae* L.) salinity stress induces responses at morphological, physiological and molecular levels, affecting plant productivity. Four olive cultivars with differential tolerance to salt were grown under saline conditions in long barrels for regular root growth to mimic field conditions. Arvanitolia and Lefkolia were previously reported as tolerant to salinity, and Koroneiki and Gaidourelia were characterized as sensitive, exhibiting a decrease in leaf length and leaf area index after 90 days of salinity. Prolyl 4-hydroxylases (P4Hs) hydroxylate cell wall glycoproteins such as arabinogalactan proteins (AGPs). The expression patterns of *P4Hs* and *AGPs* under saline conditions showed cultivar-dependent differences in leaves and roots. In the tolerant cultivars, no changes in *Oe*P4H and *Oe*AGP mRNAs were observed, while in the sensitive cultivars, the majority of *Oe*P4Hs and *Oe*AGPs were upregulated in leaves. Immunodetection showed that the AGP signal intensity and the cortical cell size, shape and intercellular spaces under saline conditions were similar to the control in Arvanitolia, while in Koroneiki, a weak AGP signal was associated with irregular cells and intercellular spaces, leading to aerenchyma formation after 45 days of NaCl treatment. Moreover, the acceleration of endodermal development and the formation of exodermal and cortical cells with thickened cell walls were observed, and an overall decrease in the abundance of cell wall homogalacturonans was detected in salt-treated roots. In conclusion, Arvanitolia and Lefkolia exhibited the highest adaptive capacity to salinity, indicating that their use as rootstocks might provide increased tolerance to irrigation with saline water.

## 1. Introduction

The olive tree is known to grow in adverse environments, ranging from dry conditions to heavy-rain areas [1]. However, the Mediterranean basin, which is the major olive cultivation region in the world, has been identified as one of the “Hot-Spots” in future climate change projections, posing a threat to olive cultivation [2].

Climate change in the Mediterranean basin will affect year-round precipitation, considering that the majority of olive trees grow under rainfed conditions [3]. In this case, low-quality saline water will be used for the irrigation of olive orchards, inducing salinity stress and limiting vegetative growth and productivity [4]. 

Specifically, it was reported that two different NaCl concentrations induced a significant decrease in the total plant and root dry mass in three different olive cultivars in the course of one year [5]. The higher-NaCl treatment reduced the plant and root dry mass by 83% in one cultivar [5]. Moreover, the yield was reduced by 30% after saline water treatment in experimental olive orchards in Spain [6,7]. Four cultivars, including Koroneiki, were exposed to 240 days of salt stress, which also resulted in a severe reduction in the total plant, root and leaf dry mass [8]. Arbequina exhibited the highest reduction in the total plant dry mass at 67.5%, while in Koroneiki, it decreased by 59.3% [8]. 

Salinity affects photosynthesis, increases photorespiration, alters cell homeostasis and causes the production of reactive oxygen species (ROS) [9]. Olive is considered a moderately salt-tolerant plant [10], which is mainly cultivar-dependent [11]. Salt tolerance in olive cultivars was associated with effective mechanisms of ion exclusion and the retention of Na+ and Cl- in the roots [12,13], along with a reduction in the water mass flow and a decrease in relative growth [12]. Specifically, the leaf area and shoot length were reduced according to measurements of fresh and dry mass [11,14].

Nine Greek olive cultivars were evaluated in response to NaCl treatment, with the results indicating that two of them, Arvanitolia and Lefkolia, showed the highest tolerance to salinity compared to the others, including Koroneiki [15,16]. These two cultivars exhibited the highest total plant dry and fresh mass as well as leaf fresh mass under saline conditions [15]. In addition, excessive leaf drop and the highest decrease in leaf dry mass were observed in Gaidourelia among five cultivars under NaCl stress, while the total plant dry mass was also significantly reduced in Koroneiki [17]. Moreover, Lefkolia showed the least severe growth inhibition symptoms associated with salinity [17]. The lowest sodium content and the highest K/Na ratio in leaves were observed in Lefkolia and Arvanitolia, while a higher sodium content and the lowest K/Na ratio were detected in Koroneiki [16]. 

The reduction in photosynthetic rates under salt stress was correlated with a decrease in chlorophyll content, which might be due to impaired biosynthesis or pigment degradation [18]. The chlorophyll content increases in salt-tolerant species, whereas it decreases in salt-sensitive plants [19,20]. Exposure to salinity induces oxidative stress, followed by ROS generation and accumulation [21], causing damage to lipids and proteins and resulting in lipid peroxidation, an indication of lipid damage [22]. The malondialdehyde (MDA) content is widely accepted as a marker of lipid peroxidation [23].

The identification of olive cultivars that are tolerant to salinity will limit the detrimental effects of irrigation with saline water on productivity. 

Salinity is considered a major threat to the cell wall structural integrity and is monitored by cell wall sensors [24]. These sensors initiate transcriptional responses that, in turn, induce the biosynthesis and deposition of lignin in secondary cell walls [25], providing mechanical support, impermeability and resistance to biodegradation [26]. The major components of pectin polysaccharides are homogalacturonans (HGs), composed of linear polymers of α-(1-4)-linked galacturonic acids that are highly methyl-esterified. De-esterified HGs form junctions with the divalent cations of Ca^2+^, leading to the formation of egg-box structures, which might be considered responsible for cell wall stiffness, softening and swelling [27].

Cell wall hydroxyproline-rich glycoproteins (HRGPs), such as arabinogalactan proteins (AGPs), were shown to be involved in salinity stress [28,29,30,31] and other abiotic stresses, such as mechanical wounding and oxygen deficiency [32]. Specifically, a significant decrease in plasma membrane AGPs was demonstrated in tobacco cell cultures under NaCl stress [29,30], while in salt-adapted tobacco BY-2 cell cultures, a reduction in AGP-bound epitopes was observed in the cytoplasm, plasma membrane and tonoplast [31]. This lower content of AGP epitopes was accompanied by high AGP accumulation in the culture medium [31]. However, a massive increase in Yariv-reagent-bound AGPs was observed in salt-stressed BY-2 tobacco cells [30]. Similarly, an increase in AGP content was detected in the xylem sap of *Brassica olarecea* in response to salinity [33]. The Arabidopsis SOS5 gene encodes Fasciclin-like AGP4 (FLA4), which causes defects under salt stress in the roots of *sos5* mutants [34,35]. Moreover, Arabidopsis knockdown mutants of hydroxyproline-O-galactosyltransferases (GALTs), which catalyze the glycosylation of AGPs, exhibited a root growth inhibition phenotype in response to NaCl treatment [36]. 

The exposure of the model grass *Βrachypodium distachyon* to salinity resulted in the induction of several *AGPs* and *FLAs*, particularly *FLA11* [37]. In addition, the immunolocalization of AGPs indicated a higher signal intensity of AGP-bound epitopes in root rhizodermal and exodermal cell walls under salt treatment [37]. In the seagrass *Zostera marina* L., unique features of the AGP glycan structure in relation to branching were determined, which might be related to calcium binding as an adaptation mechanism to saline environments [38]. Moreover, the mutation of AGP fucosyltransferases in Arabidopsis increased the sensitivity of roots to salinity, inducing a shorter root phenotype than that of wild-type plants, indicating the involvement of fucosylation in cell expansion [39]. 

Genes involved in the glycosylation of AGPs might also be considered regulatory components of the content and function of AGPs. It was reported that the genome editing of one or two GALT genes resulted in a reduction in Arabidopsis plant growth, indicating the importance of AGP glycosylation for proper function [40,41]. AGPs are also hydroxylated at proline residues by prolyl 4-hydroxylases (P4Hs) and are subsequently *O*-glycosylated [42,43,44], indicating that alterations in the expression of P4Hs result in alterations in the content of AGP-bound epitopes [45]. Therefore, GALTs and P4Hs might also be involved in the salinity response through their regulatory role in AGP glycosylation.

In this study, the comparative responses of cell wall glycoproteins, part of their biosynthetic machinery and cell wall pectins were investigated under long-term salinity stress in four olive cultivars with differential tolerance to this stress. The objective was to determine the involvement of these cell wall components in the molecular basis of tolerance to salinity stress. The morphological and physiological parameters, as well as the gene expression patterns of olive P4Hs and AGPs, were determined in the four olive cultivars. Moreover, the AGP accumulation levels, immunolocalization patterns and the levels of methyl- and demethyl-HGs in roots indicated differential responses at the transcript, protein and pectin levels depending on the cultivar’s salt tolerance levels. Significant root cell structure alterations, such as cell wall thickening and changes in the cortical cell size, were observed in tolerant and sensitive cultivars. Moreover, aerenchyma formation was observed after 45 days of treatment in the salt-sensitive cultivar.

## 2. Materials and Methods

### 2.1. Plant Material

Two-year-old olive trees of Koroneiki, Arvanitolia, Lefkolia and Gaidourelia were placed in containers with a 150 L volume and 90 cm length containing approximately 130 lt of a sand–perlite mixture with 50% sand and 50% perlite and were fertilized with half-strength Hoagland solution (Appendix A) [46,47]. The olive trees were grown for 10 months until the rooting system was fully developed, as is shown for Koroneiki in Figure 1A. The containers were placed one meter apart from each other, while the cultivars were positioned randomly in four rows. Ten trees per cultivar were used for the experiments: five trees for the salinity condition and five for the control. Fertilization was performed either manually or through a hydroponic system via Hoagland nutrient solution. Full-strength The NaCl concentration was set at 120 mM, and the application frequency was constant in every irrigation. The experiment was conducted for a period of 90 days starting in spring. At least three biological replicates from roots and leaves were collected at 0 days and after 45 and 90 days of NaCl treatment. Roots and fully developed leaves were ground into fine powder in liquid N_2_ and stored at −80 °C. 

### 2.2. Morphological Analysis of Leaves

Fully developed leaves were collected from the middle parts of newly grown shoots around the tree at a height similar to the shoulder level of an average human. After scanning the leaves at a 600 dpi resolution, they were subjected to segmentation by using an algorithm [48] implemented in MATLAB (The Mathworks Inc., Natick, MA, USA) using the Image Processing Toolbox. This method was used for object contour extraction from binary images and identified various geometrical characteristics, which were assigned to different morphological traits. 

### 2.3. Chlorophyll Pigment and MDA (Malondialdehyde) Determination

Fully developed leaf tissues were collected at least in triplicate from the salinity treatment time course points of the four cultivars and were used to extract chlorophyll pigments according to the DMSO-based protocol [49]. Chlorophyll a (Chla), chlorophyll b (Chlb) and total chlorophyll (Chlt) were determined using Arnon’s (1949) equations. Lipid peroxidation was determined by using the TBA (thiobarbituric acid) reaction, and then the MDA content was quantified [50,51].

### 2.4. Protein Prediction of AGPs and P4Hs and Phylogenetic Analysis of P4Hs

We developed an algorithm in MATLAB (The Mathworks Inc., Natick, MA, USA) to identify possible classical AGPs from the Olea europaea proteome using the bioinformatics method in [52]. Classical AGPs were initially discriminated for biased amino acid compositions that were at least 50%. The identified proteins were then further investigated with criteria regarding the existence of a signal peptide, as well as the presence and placement of repeats of the dipeptides AP, PA, SP and TP, as these sequences are frequently seen in known AGPs [52]. 

Nineteen previously classified Arabidopsis *P4H* genes were used to generate a phylogenetic tree comprising the olive, tomato and grape *P4H* genes. The deduced amino acid sequences of the OeP4Hs were aligned using MUSCLE 3.8 (EMBL-EBI, Hinxton, UK) [53], employing default settings. Positions with gaps of ≥20% were removed using GUIDANCE2 (Tel-Aviv University, Tel-Aviv, Israel) [54]. In silico analysis defined the gene structure and domain organization. Specifically, the N-terminal signal peptide positions were detected using Phobius (EMBL-EBI, Hinxton, UK) [53], WOLF Psort (NAKAI Lab, Tokyo, Japan) [55] and signal IP 5.0 (DTU, Kongens Lyngby, Denmark) [56], and the transmembrane helices and ER membrane retention signals were identified by TMHMM 2.0 (DTU, Kongens Lyngby, Denmark) [57], TMpred (bio.tools, DTU, Kongens Lyngby, Denmark) [58] and WOLF Psort. NetNGlyc 1.0 (DTU, Kongens Lyngby, Denmark) [59] was applied to detect the N-glycosylation sites, while the cytoplasmic tails were predicted via WOLFPsort. The 3D structure of the OeP4H1 protein was performed using SWISSMODEL [60]. 

### 2.5. RNA Extraction and Real-Time RT-PCR

Total RNA was isolated from mature roots in a non-destructive way, considering that the trees were grown in 90 cm long containers, using the PureLinkTM RNA Mini Kit (Thermo Fisher Scientific^®^ Waltham, MA, USA). Total RNA was isolated from mature leaves by using a Phenol–Chloroform protocol [61,62]. Total RNA was further purified with NucleoSpin^®^ RNA Clean-up XS (Macherey-Nagel, Duren, Germany). Total RNA was incubated with Dnase I Rnase-free enzyme (Thermo Scientific, Waltham, MA, USA) and then fractionated on an RNA-denaturing GelRed^®^ (Biotium, Fremont, CA, USA)-stained 1.5% agarose gel by electrophoresis. Approximately 1–2 μg of total RNA was reverse-transcribed into first-strand cDNA with SuperScript™ II RT (Invitrogen,Waltham, MA, USA). Real-time RT-PCR was conducted on the CFX Connect™ Real-Time PCR Detection System (Bio-Rad^®^, Hercules, CA, USA) with SYBR Select Master Mix (Applied Biosystems^®^, Waltham, MA, USA). The data were analyzed by using the 2−Δ(ΔCt) method [63,64]. The olive Act7a Actin cDNA (identifier: OLEEUCl025648|Contig2 OLEAEST TC) was used as a reference gene [65]. The primers for the gene expression studies are presented in Appendix A. Three biological replicates were used for each sample.

### 2.6. Western Blotting 

Total proteins were extracted according to Woodson [66] with some modifications. Approximately 0.2 g of root and leaf tissue was ground in liquid nitrogen, and then extraction buffer (65 mM Tris HCl, pH 6.8, 2% (*w*/*v*) SDS, 700 mM β-mercaptoethanol, 2 mM EDTA, 1:10 volume protease inhibitor and 1 mM PMSF) was added at a ratio of 1:3 volume. After vortexing, the mixture was boiled at 99 °C for 5 min and centrifuged for 10 min at 4 °C. Approximately 20 μg protein samples, after Bradford assay quantification, were separated by SDS-PAGE in a 10% polyacrylamide gel (30% acrylamide/bisacrylamide) and transferred to a hydrophobic PVDF membrane (Millipore Immobilon-P, Burlington, MA, USA). The membrane was blocked with 5% non-fat dry milk dissolved in 1 × TBST buffer for 1 h at room temperature. The LM2 and JIM13 antibodies (Plant Probes, Leeds, UK) were used to determine epitope-bound AGPs. The β-actin monoclonal antibody (ANTIBODIES, Germany) was used as a loading control. The secondary antibodies that were used were an anti-rat monoclonal antibody (Agrisera, Geneva, Switzerland) for LM2 and JIM13 and an anti-mouse monoclonal antibody (Agrisera, Switzerland) for β-actin. A chemiluminescent solution (SuperSignal West Pico Chemiluminescent Substrate, Thermo Scientific, Waltham, MA, USA) was used for detection. After incubation, the membranes were exposed to X-ray film for 1–2 min. 

### 2.7. Root Fixation

Whole root parts, 1 cm in length, were fixed for 1.5 h in 4% (*w*/*v*) paraformaldehyde in PEM buffer (50 mM piperazine-1,4-bis (2-ethanesulfonic acid, 5 mM ethyleneglycol-O,O′-bis(2-aminoethyl)-N,N,N′,N′-tetraacetic acid, 5 mM MgSO_4_·7H_2_O, pH 6.8) containing 2% (*v*/*v*) dimethyl sulfoxide and 0.1% (*v*/*v*) Triton X-100. The specimens were dehydrated in a graded ethanol series (10–90%) diluted in distilled water and finally three times in absolute ethanol for 30 min (each step) on ice. The material was post-fixed with 0.25% (*w*/*v*) osmium tetroxide added to the 30% ethanol step overnight at 4 °C during the dehydration process. The material was infiltrated with LR White (LRW; Sigma, Darmstadt, Germany) acrylic resin diluted in ethanol in 10% steps (10% LRW in ethanol, followed by 20% and gradually replaced by higher LRW concentrations until the total replacement of ethanol with 100% LRW (1 h in each step) at 4 °C. When pure resin was finally applied, the samples were embedded in gelatin capsules filled with LRW resin and polymerized at 60 °C for 48 h. 

### 2.8. Root Sectioning by Ultramicrotome and Toluidine Blue O Staining

After they were embedded in LRW, the samples were sectioned using an ULTROTOME III TYPE 8801A ultramicrotome (LKB, Stockholm, Sweden) equipped with a glass knife. Semithin root sections (0.5–2 μm) were placed on a microscope slide and stained with 0.5% (*w*/*v*) toluidine blue O to observe the morphology and the structure of the roots.

### 2.9. Callose and Lignin Localization

Callose was localized in root semithin sections (0.5–2 μm) stained with 0.05% (*w*/*v*) aniline blue (Sigma, C.I 42725) in 0.07 M K_2_HPO_4_ buffer, pH 8.5. Sections remained in aniline blue solution during observation with a Zeiss Axioplan microscope (Zeiss Oberkochen, Germany) equipped with a micrometric scale, a differential interference contrast (DIC) system and an Axiocam MRc5 digital camera. Lignin was observed under the optical microscope after staining with 3% Phloroglucinol. 

### 2.10. Cell Wall Epitope Immunolocalization

For cell wall epitope immunolocalization, the protocol described by Giannoutsou et al. [67] was applied. In detail, root semithin sections were blocked in 5% bovine serum albumin (BSA) for about 1.5–2 h. Then, the samples were thoroughly washed with pH 7 phosphate-buffered saline (PBS) (3 × 10 min), followed by an overnight incubation with the appropriate primary antibody diluted 1:40 in PBS (pH 7) at room temperature. After being rinsed with PBS (pH 7/3 × 10 min), the samples were incubated with the appropriate secondary antibody diluted 1:40 in PBS (pH 7) for 3 h at 37 °C. Finally, the samples were once again rinsed three times with PBS (pH 7) for 10 min and mounted with a mixture of glycerol/PBS (2:1, *v*/*v*) containing 0.5% (*w*/*v*) p-phenylenediamine (anti-fade solution). 

For the immunostaining of HGs, the protocol described by Giannoutsou et al. [67] was applied. LM-20, LM-19 and LM-18, as well as JIM-7 and JIM-5 (Plant Probes, Leeds, UK), were used as primary antibodies, and FITC-conjugated anti-rat IgG (Sigma, Darmstadt, Germany) was used as the secondary antibody. All antibodies were diluted 1:40 in PBS at pH 7. During the immunolabeling procedure, the sections were washed with PBS. LM-20 is a monoclonal antibody that recognizes the HG domain of pectic polysaccharides. The antibody requires methyl-esters for the recognition of HGs and has no known cross-reactivity with other polymers. It does not bind to unesterified HGs and was applied for the detection of fully methyl-esterified pectin. JIM-7 recognizes partially methyl-esterified epitopes of HGs having a high degree of methyl esterification. LM-20 was recommended as a cross-check in place of JIM-7 due to its higher specificity for fully methyl-esterified HGs. JIM-5 is capable of binding even to unesterified HGs, something that JIM-7 is incapable of. The LM-19 monoclonal antibody was also applied for the detection of demethyl-esterified HGs but appears to prefer unesterified HGs. This antibody was also used as a cross-check in place of JIM-5 due to its ability to bind more efficiently to unesterified HGs. Lastly, the LM-18 monoclonal antibody binds to partially methyl-esterified HGs (Plant Probes, Leeds UK) (Appendix A). 

For the immunostaining of arabinogalactan proteins, JIM13 (Plant Probes, Leeds, UK) was used as the primary antibody, and FITC-conjugated anti-rat IgG (Sigma, Darmstadt, Germany) was the secondary antibody. All antibodies were diluted 1:40 in PBS with a pH of 7. During the immunolabeling procedure with the antibodies, the sections were washed with PBS (pH 7).

## 3. Results

### 3.1. Leaf Morphological Characteristics in Response to Salinity Stress

By using the measuring tools in ImageJ (NIH, Bethesda, MD, USA) [68], we measured the length of the depicted root at 86.783 cm, while the convex hull area was estimated at 2825.017 cm^2^ (Figure 1B). The salt treatment comprised 120 mM NaCl for a 90-day time course, while root and leaf tissues were sampled after 45 and 90 days (Figure 1). 

Significant variation in morphological responses was observed among cultivars. Koroneiki showed smaller leaves (Figure 1C) and necrosis at the edge of fully developed leaves, accompanied by severe leaf drop after 90 days, as was previously reported [69]. The leaf area index of NaCl-treated Koroneiki decreased by approximately 30% after 45 days and an additional 10% by the end of 90 days (Figure 1D). 

The leaf area Index of Arvanitolia decreased by almost 20% in 45 days of salinity and fully recovered to pre-treatment levels after 90 days (Figure 1D). Decreases in the leaf height of almost 20% and 8% were observed in Koroneiki and Gaidourelia, respectively (Figure 1D).

The maximum transversal leaf diameter (the greatest horizontal diameter of the leaf blade) in Arvanitolia decreased after 45 days, while it had recovered to pre-treatment levels by 90 days. However, in Koroneiki, a salt-sensitive cultivar, the maximum transversal leaf diameter decreased by 10% and 15% after 45 and 90 days, respectively (Figure 1D).

### 3.2. Leaf Chlorophyll and Malondialdehyde (MDA) Contents under Saline Conditions

The leaf Chla and Chlb contents were determined after 45 and 90 days of salinity (Figure 2A). In salt-sensitive Koroneiki, Chla and total chlorophyll (Chlt) decreased after 45 days, while no differences in Chlb were observed (Figure 2A). However, in Arvanitolia, Chla, Chlb and Chlt increased, while in Lefkolia, only Chlb increased after 45 days (Figure 2A). No alterations in the contents of Chla, Chlb or Chlt were observed in Gaidourelia (Figure 2A). 

The content of Chla was higher than that of Chlb in all cultivars in the control and salinity treatments (Figure 2A). However, the Chla-to-Chlb ratio showed significant variation between salt-tolerant and salt-sensitive cultivars (Table 1). In Koroneiki, the Chla/Chlb ratio decreased from 2.5 to 1.6 and from 2.3 to 1.1 after 45 and 90 days, respectively (Table 1). Approximately similar reductions in the Chla/Chlb ratio were also observed for Gaidourelia (Table 1). Salt-tolerant cultivars, Arvanitolia and Lefkolia, exhibited slightly higher Chla/Chlb ratios after 95 days of salinity (Table 1). 

The lipid peroxidation of membranes was also determined by the content of malondialdehyde (MDA) accumulation in the leaf tissue of all four cultivars (Figure 2B). The MDA content significantly increased in Koroneiki and Gaidourelia after 45 and 90 days of salinity, respectively (Figure 2B). The salt-tolerant cultivars, Arvanitolia and Lefkolia, did not show any changes in the MDA accumulation rate, possibly indicating a lack of oxidative stress (Figure 2B). 

### 3.3. Olive P4H Gene Family

A phylogenetic tree comprising 16 olive genes was constructed based on 19 previously classified Arabidopsis genes (Figure 3; personal communication). Each cluster comprised an Arabidopsis gene, which was considered the basis, as well as olive, tomato and grape genes with the highest identity at the amino acid level. Multiple alignment was performed, and all the important motifs and essential amino acids for P4H activity are highlighted (Appendix A). 

The deduced olive, tomato and grape P4H amino acid sequences were grouped into clusters based on their percent identity with Arabidopsis P4Hs (Figure 3). The basis for each cluster was an Arabidopsis polypeptide. The 16 putative OeP4H polypeptides were clustered into six groups (Figure 3). Several deduced olive P4H polypeptides also showed high percent identity among them, which ranged from 76 to 84% and were considered olive P4H-like polypeptides. *Oe*P4H3, *Oe*P4H10, *Oe*P4H1 and *Oe*P4H9 showed 78%, 76%, 83%, 84% and 79% identity with the *Oe*P4H3-like, *Oe*P4H10-like, *Oe*P4H1-like, *Oe*P4H9-like1 and *Oe*P4H9-like2 polypeptides, respectively.

The more divergent group in terms of percent identity at the amino acid level comprised four P4Hs, *Oe*P4Hs, *Oe*P4H2, *Oe*P4H6, *Oe*P4H7 and *Oe*P4H12, while the group also included four, *Oe*P4H3, *Oe*P4H3-like, *Oe*P4H10 and *Oe*P4H10-like, with a high percentage of identity at the amino acid level (Figure 3). *Oe*P4H9, *Oe*P4H9-like 1 and *Oe*P4H9-like 2 were members of a distinct group (Figure 3). One cluster comprised only one olive P4H, *Oe*P4H14, while another two clusters included two olive P4Hs each: *Oe*P4H15 and *Oe*P4H15-like in one cluster and *Oe*P4H1 and *Oe*P4H1-like in the other (Figure 3).

The gene structure and domain organization were conserved in olive P4Hs according to an in silico analysis (Appendix A). The positions of the exons for each gene and the positions of the protein domains in the deduced amino acid sequence of each P4H polypeptide were conserved. The P4Hc (IPR006620, SM00702) and Fe_2_OG-Oxy (IPR005123, PF13640) domains are located at the C-terminal region of the predicted *Oe*P4H protein sequences (Appendix A). 

P4Hs are type II membrane-anchored proteins localized in the ER and Golgi. Almost all of the OeP4Hs comprise transmembrane helices or an ER retention signal at the N-terminal or C-terminal region or both (Appendix A). The ER retention signal might indicate permanent localization in the ER, but the presence of a cytoplasmic tail facilitates its transport to the Golgi via the secretory pathway. The cytoplasmic tail comprises 5–20 amino acids (Appendix A) and is located in the N-terminus of the protein. The cytoplasmic tail consists of positively charged amino acids similar to the RXR motif or the dibasic signal found in mammalian and yeast Endoplasmic Reticulum (ER) and Golgi proteins. Similar cytoplasmic domains have been found to be responsible for the transfer of prolyl 4 hydroxylases to the Golgi apparatus in plants. Mutant GFP prolyl 4 hydroxylase proteins in which the basic amino acids of the cytoplasmic tail were substituted with non-charged hydrophilic amino acids were found to be located in the ER in tobacco BY-2 cells, unlike the control proteins, which were observed in both the ER and Golgi apparatus [70].

Signal peptide and *N*-glycosylation prediction tools indicated that 7 out of the 16 *Oe*P4Hs were predicted to have a signal peptide as well as *N*-glycosylation sites, which indicated that these proteins might undergo post-translational modifications. Based on the 3D structure of *Oe*P4H (Appendix A), 46 residues comprising the signal peptide and the transmembrane domain were missing. 

### 3.4. Gene Expression of P4Hs and AGPs in Roots and Leaves in Response to Salinity Stress

Despite the involvement of plant P4Hs in abiotic stresses, no information on their response to salinity is available for any plant species, including olive. Among the 16 putative olive P4Hs, only 7 showed detectable expression levels in either roots or leaves (Figure 4). 

In roots, the transcript levels of most *Oe*P4Hs were stable under saline conditions in tolerant Arvanitolia among all cultivars (Figure 4). Expression peaks for *OeP4H2* and *OeP4H1-like* were observed at 45 days, while *OeP4H7* was downregulated after 90 days in roots as well as in leaves (Figure 4). *OeP4H10-like* was downregulated in leaves after 90 days (Figure 4).

Salinity-tolerant Lefkolia exhibited expression peaks for *OeP4H1*, *OeP4H2*, *OeP4H6* and *OeP4H9* in roots after 45 days, while *OeP4H6* and *OeP4H9* were upregulated and *OeP4H7* was downregulated at 90 days (Figure 4). In leaves, the transcript abundance of *OeP4H9*, *OeP4H6*, *OeP4H2* and *OeP4H10-like* decreased at 90 days, while that of *OeP4H7* increased (Figure 4). Moreover, the expression of *OeP4H1-like* was upregulated at both time points in leaves, and that of *OeP4H1* increased at 45 days and decreased at 90 days (Figure 5).

In the salt-sensitive cultivar Koroneiki, the expression patterns of *OeP4H2* and *OeP4H9* in roots increased at 90 days, while *OeP4H1-like*, *OeP4H10-like*, *OeP4H7* and *OeP4H1* were downregulated (Figure 4). In leaves, *OeP4H1*, *OeP4H2*, *OeP4H9* and *OeP4H10-like* were upregulated at both time points, while *OeP4H6* and *OeP4H1-like* were upregulated only after 45 days of salinity stress (Figure 4).

In Gaidourelia, a salt-sensitive cultivar, *OeP4H1*, *OeP4H6* and *OeP4H9* showed expression peaks at 45 days in roots, and *OeP4H10-like* had an expression peak only at 90 days (Figure 4 and Figure 5). The transcripts of both *OeP4H1* and *OeP4H1-like* decreased after 90 days (Figure 4 and Figure 5). In leaves, the *OeP4H1* and *OeP4H6* expression levels were upregulated after 90 days, while those of *OeP4H9* and *OeP4H10-like* increased after both 45 and 90 days and after 45 days only, respectively (Figure 5).

The expression levels of two AGPs, *OeAGP4-like* and *OeAGP10-like*, were determined in roots and leaves during the salinity treatment time course (Figure 5). In Arvanitolia, *OeAGP4-like* and *OeAGP10-like* exhibited no changes in expression in roots, while in leaves, *OeAGP4-like* was down- and upregulated after 45 and 90 days, respectively (Figure 5). Lefkolia showed an increase in transcript abundance for *OeAGP10-like* in roots and the upregulation of both AGPs in leaves throughout the salinity treatment time course (Figure 5). In Koroneiki, the transcript levels of *OeAGP4-like* and *OeAGP10-like* were downregulated in roots: the first after 45 and 90 days and the second after 90 days, respectively (Figure 5). In leaves, a decrease in expression was observed for *OeAGP4-like* and *OeAGP10-like* after 45 days, which was followed by upregulation after 90 days only for *OeAGP10-like* (Figure 5). For Gaidourelia, *OeAGP4-like* showed a significant increase in transcript levels after 90 days in roots, while *OeAGP10-like* was up- and downregulated after 45 and 90 days, respectively (Figure 5). In leaves, *OeAGP4-like* transcripts decreased after 45 days, while those of *OeAGP10-like* were down- and upregulated after 45 and 90 days, respectively (Figure 5). No trends in AGP expression among the four cultivars were observed, despite minor changes in transcript abundance throughout the salinity treatment time course.

### 3.5. Protein Levels of AGP-Bound Epitopes in Roots and Leaves in Response to Salinity Stress

The AGP-bound epitopes were detected by Western blot analysis in roots in response to salinity (Figure 6). LM2-bound AGPs in Koroneiki, Gaidourelia and Lefkolia showed stable protein levels after 45 and 90 days of exposure to salinity stress, while, in Arvanitolia, an increase was observed mainly after 90 days of treatment (Figure 6). The JIM13-bound AGPs also exhibited stable accumulation levels in Koroneiki, Arvanitolia and Lefkolia under saline conditions, while a marginally lower signal was observed in Gaidourelia at both time points (Figure 6).

### 3.6. Cell Morphology, AGPs and Pectin Immunolocalization in Olive Roots

#### 3.6.1. Cell Morphology and AGP and Pectin Immunolocalization in Koroneiki Roots

The effect of salinity on the anatomy and cell morphology in Koroneiki was determined by analyzing root cross-sections (Figure 7A–K). The outer cell layer constitutes the rhizodermis (1 in Figure 7D,F), while multiple parenchyma cell layers form the cortex (2b in Figure 7B,F), in which the exodermis is the first layer (2a in Figure 7D,F,I) and the endodermis is the innermost layer (2c in Figure 7B,G–I). The vascular tissue is surrounded by the pericycle (3c in Figure 7C,G–I), while a distinct separation of xylem (3a in Figure 7C,G,I) and phloem elements (3b in Figure 7C,G,I) was visible.

The cortical cells of the control root were round and arranged in homocentric circles (Figure 7A–D). The cortical cells proximal to the endodermis were smaller and compactly arranged without intercellular spaces, while those near the exodermis were larger with small and equally distributed intercellular spaces (Figure 7B–D). Differences in the morphology of specific cell types in root sections were observed under saline conditions. In the salt-treated cortex, the cells were arranged in a non-orderly way, and the intercellular spaces were variable, ranging in area from small to large, indicating the formation of aerenchyma, which is usually observed under conditions of oxygen deficiency, such as submergence and waterlogging (arrows in Figure 7I–K). The diameters of cells ranged from 10 μm to more than 20 μm (Figure 7F–K), while the shapes were circular, oval, elongated and, in some cases, polygonal (Figure 7F–K). The endodermal cells were compactly arranged without intercellular spaces, despite their size heterogeneity (arrows in Figure 7L). The phloem elements were distinct between the xylem rays (3b in Figure 7L,M).

In the control roots, after aniline blue staining, the xylem of the vascular cylinder fluoresced (3a in Figure 7L), as did the sclerenchymatic tissue surrounding the sieve tubes and companion cells of the phloem (3b in Figure 7L). In the control roots, the cell walls of the endodermis were not thickened, with only a weak fluorescent signal at the anticlinal cell walls (arrows in Figure 7L). In contrast, in the salt-treated roots, the thickening in the endodermal cells appeared not only at the anticlinal but also at the periclinal cell walls (arrows in Figure 7M). Changes were also observed in the exodermis of the control and salt-treated samples. The cells of the exodermis were larger than those observed in the control sample (compare 2a in Figure 7O to 2a in Figure 7N), and their cell walls fluoresced intensely after staining with aniline blue (2a in Figure 7O). 

JIM13-bound AGPs were localized throughout the root cross-sections (Figure 7P,Q), while a weaker fluorescent signal was observed under saline conditions, especially in the cortex (Figure 7R,S). Fully methyl-esterified HGs detected by the LM20 antibody were present in the cortical cells, as well as in the vascular cylinder in control root cross-sections (Figure 8A,C). A distinct signal was observed in the younger phloem cell walls near the cambium (Figure 8C), while the anticlinal cell walls of the endodermis exhibited a lack of an HG localization signal, possibly due to the presence of the Casparian strip formed at the anticlinal cell wall (arrow in Figure 8C). In the salt-treated roots, the cortical cells showed a weaker signal (Figure 8D), as did the periclinal endodermal cell walls, but not the anticlinal ones (arrow in Figure 8E).

The JIM7 antibody, which can bind to partly demethyl-esterified HGs that display a high degree of methyl esterification, showed a lower fluorescent signal in the root sections of the salinity treatment in comparison to the control (Figure 8F–H). In the salt-treated samples, deformations in the shape of cortical cells appeared (Figure 8G,H), as well as large intercellular spaces (arrows in Figure 8G,H). 

JIM5-HG epitopes were detected throughout almost the entire salt-treated and control root sections (Figure 8I–K). Demethyl-esterified homogalacturonans were detected in younger phloem cell walls closer to the cambium, as well as in the cortical cell walls (Figure 8I). Demethyl-esterified HGs were not detected in the anticlinal endodermal cell walls (arrows in Figure 8I). Although JIM5 fluoresced evenly in the cortical cells of the control sample (Figure 8J), in the salt-treated root samples, the cells displayed some acute signaling spots at the junction sites between two neighboring cells (Figure 8K). LM18 immunolocalization resulted in similar patterns in control and salt-treated roots (Figure 8L–Q). Cortical cell deformities were observed in salt-treated root cross-sections (Figure 8M,P,Q) compared to control ones (Figure 8L,O). The vascular cylinder phloem cells exhibited fluorescent signals in control and salt-treated roots (Figure 8N,Q). 

#### 3.6.2. Cell Morphology and AGP and HG Immunolocalization in Lefkolia Roots 

The control cortical cells of Lefkolia roots are similar in shape, arrangement (Figure 9A–C) and size distribution (2b and 3c in Figure 9B,C) to those of Koroneiki. The salt-treated cortical cells were compactly arranged, but their shape was irregular and variable, while intercellular spaces also varied in size (Figure 9D–F). The salt-treated Lefkolia endodermal cells were distinct from the cortex and pericycle (3c in Figure 9F), because they were nicely arranged in homocentric cycles, unlike in Koroneiki. The xylem and phloem elements were easily recognizable (3a and 3b in Figure 9F). Although salt-treated Lefkolia roots exhibited diversity in cortical cell morphology, no cell wall deformations were observed, while the cell shape was more regular than in Koroneiki. Moreover, the formed intercellular spaces were larger in Koroneiki. Overall, the salt-induced cell morphology changes in Lefkolia were minor, indicating a better adaptation capacity. 

Aniline blue revealed a slight thickening of control exodermal cells (Figure 9G), while cells of the endodermis fluoresced at their anticlinal cell walls (Figure 9G). In the salt-treated samples, thickening occurred not only in the cell walls of the exodermis but also in 2–3 rows of cortical cells below the exodermis (2b in Figure 9H). The anticlinal cell walls of the endodermis of the salt-treated samples were intensely thickened (2c in Figure 9I). 

The JIM13-bound AGPs displayed a strong fluorescence signal only in the rhizodermis of the control roots (Figure 9J) but not in salt-treated roots (Figure 9K). Intense AGP signaling was detected in salt-treated cortical cells (Figure 9L). A strong AGP signal was observed in the control xylem but not in the phloem (Figure 9J), while an AGP signal was detected in both the salt-treated root phloem and xylem (Figure 9L).

The control root exhibited an intense LM20-HG epitope signal in phloem cell walls (Figure 10A), while, in salt-treated roots, the signal was detected mostly in the sclerenchymatic fibers (Figure 10B). Fully methyl-esterified homogalacturonans were present throughout the cortex in control and salt-treated roots (Figure 10A and Figure 10B, respectively). No signal was observed in the endodermal anticlinal cell walls of the control and salt-treated samples (arrows in Figure 10C and Figure 10D, respectively). The absence of fully methyl-esterified HGs at the anticlinal cell walls of the endodermis is consistent with the endodermis of Koroneiki, indicating the presence of the Casparian strip at the anticlinal cell walls (arrows in Figure 10C and Figure 10D respectively). 

JIM7-HG epitopes were detected in the entire section of control and salt-treated roots (Figure 10E,F). JIM7-HG epitopes were not detected in the cell walls of the rhizodermis in either control or salt-treated samples (Figure 10E and Figure 10F, respectively). The cortical and endodermal cells of the salt-treated samples fluoresced intensely (Figure 10H), while the vascular cylinders of both control and salt-treated samples displayed a plethora of demethyl-esterified HGs with a high degree of methyl esterification (Figure 10G,H). Signals were detected in the sclerenchymatic fibers above the phloem and in young phloem cells near the cambium (Figure 10H). Partially methyl-esterified homogalacturonans were also observed in the anticlinal cell walls of the endodermis under saline conditions (arrows in Figure 10H). 

LM19- and LM18-HG epitopes were detected in control and salt-treated roots (Figure 10I–T). An unesterified homogalacturonan signal was present in the cells of the rhizodermis of the control sample but not in the salt-treated roots of Lefkolia (Figure 10I,J). Unesterified homogalacturonans were observed in the entire cortex as well as in the vascular cylinder (Figure 10I–N). LM19 epitopes were not present at the anticlinal cell walls of the control and salt-treated endodermis (arrows in Figure 10K,L). Intense signals were also observed in the sclerenchymatic fibers of the phloem in the vascular cylinder in control and salt-treated samples (arrows in Figure 10M,N), whereas younger phloem cells near the cambium showed weaker fluorescence (Figure 10M,N). Partially methyl-esterified homogalacturonans were detected throughout the entire control root structure (Figure 10O). Strong signals of partially methyl-esterified homogalacturonans, detected by the LM18 antibody, were exhibited in the phloem of control and salt-treated samples (Figure 10O,P), while no LM18 signal was detected at the cell walls of exodermal cells (Figure 10O,P). Demethyl-esterified HGs detected by the LM18 antibody were present in the cells of the rhizodermis of the control sample but not in the salt-treated roots of Lefkolia (Figure 10O,P). The lack of an LM18-HG signal in the anticlinal cell walls of the endodermis was observed again (arrows in Figure 10Q,R,S), indicating the presence of a modified cell wall at the Casparian strip site. In the salt-treated samples, no signal of the LM18-HG epitope was detected at the anticlinal cell walls of the exodermis (arrows in Figure 10T).

#### 3.6.3. Cell Morphology and AGP and HG Immunolocalization in Arvanitolia Roots

The control and salt-treated cortical cells were uniformly arranged around the root vascular cylinder (Figure 11A–F). The intercellular spaces were equally distributed between cortical cells, while the endodermal cells (3c in Figure 11C,F) surrounded the vascular cylinder, where xylem rays and phloem cells are noticed (3a and 3b in Figure 11C,F). Salt treatment did not impact either the cell morphology or root structure. No large intercellular spaces and irregular cell walls or cell wall deformities were observed. The cortical cell size and shape were regular and similar to those of the control. Interestingly, the salt-treated rhizodermal cell shape was irregular and different from that observed in the control sample (compare 1 in Figure 11D to 1 in Figure 11B).

Although Arvanitolia exhibited strongly thickened exodermal cell walls of the cortex regardless of salinity (Figure 11I,J), exodermal cells as well as the second layer of cortical cells displayed strongly thickened cell walls in response to salt treatment (Figure 11I,J). In the control roots of Arvanitolia, staining with aniline blue revealed a strong signal at the xylem of the central cylinder (Figure 11G), while the phloem did not fluoresce. Intense signals were observed at the anticlinal and, in some cells, periclinal cell walls of the endodermis (arrows in Figure 11G,H). In the control samples, Arvanitolia exhibited strongly thickened exodermal cell walls of the cortex (Figure 11I), while in salt-treated samples, not only the exodermal cells but also 2–3 layers of cortical cells were thickened, as was observed in Lefkolia samples (compare Figure 11J to Figure 9H). 

The AGP-bound epitopes exhibited a strong signal throughout the entire control root (Figure 11K). Intense signals were detected in the rhizodermis and the entire root vascular cylinder (phloem and xylem) and in cortical cells (Figure 11K). Under saline conditions, the overall fluorescent signal displayed a pattern of distribution that was similar to that in the control roots (Figure 11L). 

In the control roots of Arvanitolia, fully methyl-esterified LM20-HG epitopes were detected in the cells of the cortex and vascular cylinder of the root (Figure 12A,C). A weaker signal was observed in the entire root structure under saline conditions (Figure 12B,D). The cortex showed an overall lower fluorescent signal (Figure 12B), while the salt-treated sample exhibited a weak signal in the rhizodermis and the exodermis and an even weaker signal in cortical cells (compare Figure 12B to Figure 12A). As far as the JIM7-HG epitope is concerned, intense signaling was detected throughout the entire root structure. The rhizodermis and cortical cells displayed intense fluorescent signals in both the control and salt-treated samples (Figure 12E,F), while the anticlinal cell walls of the exodermis lacked JIM7-HG epitopes (arrows in Figure 12E,F). However, the endodermal anticlinal cell walls displayed a fluorescent signal (Figure 12E,F), as observed in Lefkolia roots too. 

LM19-HG epitopes showed similar signal intensities in control and salt-treated roots of the Arvanitolia variety. The signal was hardly detected in the cells of the exodermis in the control and the salt-treated samples (arrows in Figure 12G,H), whereas phloem and xylem elements displayed weak fluorescent signals under salt treatment (Figure 12H). The control, though, demonstrated signals in the rhizodermis and phloem (Figure 12G). 

JIM5-HG epitopes showed intense fluorescent signals in the cells of the rhizodermis of control and salt-treated roots (Figure 12I,J). In the cortical cells of the control root samples, the JIM5-HG epitope was evenly distributed at the cell walls, while in the salt-treated root samples, the JIM5-HG epitope was specifically present at the junction sites of the intercellular spaces between adjacent cells (compare Figure 12J to Figure 12I). The absence of partially methyl-esterified homogalacturonans was observed both in the anticlinal cell walls of the exodermis (arrows in Figure 12I,J) and in the endodermis of the control and salt-treated root samples (asterisks in Figure 12I,J). Weak fluorescent signals in phloem and xylem elements were detected in control and salt-treated roots (Figure 12I,J).

The cell morphology and root structure were the least affected by salt treatment, primarily in Arvanitolia and then in Lefkolia, suggesting that tolerance to salinity is associated with root morphology. Moreover, Koroneiki, a cultivar known to be sensitive to salt stress, was severely affected not only in its root structure but also in its cell wall composition. 

## 4. Discussion

Two-year-old olive trees of four cultivars were grown in containers 90 cm in height in order to mimic the growth of the rooting system in conditions similar to those in the field. The root length of a representative Koroneiki tree was 86.7 cm long, which is similar to the average root length of an olive tree growing in the orchard. In this way, the investigation of molecular adaptation under NaCl stress provided a better estimation of the cultivar response capacity in the field. 

The effect of salinity on olive trees has been thoroughly investigated and has been found to result mainly in plant growth inhibition [16,17]. The morphological characterization of the four cultivars suggested phenotypic differences in the upper part of the tree, such as significant reductions in leaf area, in accordance with previous reports [15,17]. Lefkolia showed minor symptoms and a leaf area identical to that of control trees, indicating high tolerance to salinity. Leaf tip necrosis and leaf drop were observed in Koroneiki, while Gaidourelia showed a significant decrease in leaf height by the end of 90 days, an indication of sensitivity to salinity, in accordance with previous reports [14,17]. In the salt-sensitive Koroneiki, chlorophyll a and total chlorophyll decreased under salt stress, while in Arvanitolia, a salt-tolerant cultivar, chlorophyll contents increased, including chlorophyll b. The salt-induced alterations in leaf chlorophyll might be attributed to the conversion of Chlb to Chla, thus resulting in higher Chla [71].

Root endodermal cells usually do not progress beyond the primary stage of Casparian strip formation during endodermal development. Although the endodermis remains in the primary stage in most eudicots, salinity accelerates endodermal development, and the formation of suberin lamellae frequently extends closer to the root tip. The cell walls of Koroneiki root endodermal cells are thickened and clearly distinguished by forming a fluorescent cylinder surrounding the vascular tissue. In Arvanitolia and Lefkolia, the acceleration of endodermal development and the formation of thickened exodermal cells were observed. Arvanitolia and Lefkolia exhibited 1–2 and 4–5 rows of cortical cells with thickened cell walls, respectively. Moreover, Lefkolia roots exhibited diversity in cortical cell morphology under saline conditions but no deformations in their shape or cell walls after 45 days of salinity. The intercellular spaces between root cortical cells in Lefkolia varied in size but were not as wide as in Koroneiki under saline conditions. Overall, significant differences were observed between control and salt-treated root sections in Lefkolia. 

Aerenchyma formation was observed in the Koroneiki root cortex, but not in Lefkolia or Arvanitolia after 45 days of salinity. Similar observations were reported in soybean plants and wheat seedlings [72,73]. Combined salinity and waterlogging stress resulted in cortical aerenchyma development, which improved salt resistance and sodium ion exclusion [74], while soybean plants exhibited lysogenic aerenchyma formation, particularly at high NaCl concentrations, for air space creation and the prevention of toxic ion uptake [72].

The cortical cells in contact with the exodermis fluoresced after aniline blue staining in Lefkolia and Arvanitolia. Salinity induced the deposition of callose in cortical cell walls in sorghum [75] and increased the callose-degrading enzymes in barley roots [76]. Osmotic stress is known to induce callose deposition in root cell walls [77]. The aniline blue staining of root exodermal cells is similar to results in salt-treated cotton seedling roots [78]. Moreover, salinity increased the deposition of suberin and lignin not only in endodermal but also in exodermal cells, converting them into cells impermeable to water [79]. 

The physiological significance of P4Hs under abiotic stress has not been thoroughly investigated, despite their demonstrated involvement in several growth and developmental programs [44,45,80,81]. Under saline conditions, the majority of olive P4Hs were downregulated in leaves in tolerant Arvanitolia and Lefkolia (Figure 4), while, in the two sensitive cultivars, Koroneiki and Gaidourelia, they were mainly upregulated after 90 days, indicating a putative inverse relation between high P4H expression levels and tolerance. Moreover, Arvanitolia showed minimal down- or upregulation in roots and leaves, suggesting that stable P4H expression might indicate higher tolerance. Overall, no distinct patterns of P4H expression were observed in roots among the four cultivars. All cultivars, except Koroneiki, showed an expression peak for *Oe*P4H1 after 45 days of stress, while *Oe*P4H2 increased in all four cultivars. Interestingly, *Oe*P4H7 was downregulated in all cultivars, with the exception of Gaidourelia, in which it was upregulated throughout the time course. *Oe*P4H1, *Oe*P4H6, *Oe*P4H9 and *Oe*P4H10-like showed opposite expression patterns in the leaves of salt-tolerant compared to salt-sensitive cultivars. Downregulation in salt-tolerant plants and upregulation in salt-sensitive ones were observed, indicating involvement in cultivar resilience. 

In tolerant cultivars, stable AGP expression was detected in roots, while in leaves, the upregulation of both AGPs was observed. In the roots of Populus (*Populus tremuloides*), the expression of 18 FLAs was detected under saline conditions, while 6 of them were significantly induced [82]. Moreover, among rice AGPs, only one was induced by salt in 7-day-old seedlings, while three and seven AGPs were significantly up- and downregulated by both salt and drought, respectively [83]. This is in agreement with the only two olive AGPs that showed detectable expression under saline conditions (Figure 5). In addition, in tomato roots, the expression of five AGPs was strongly repressed under saline conditions [84]. A reduction in AGP epitopes in the cytoplasm, plasma membrane and tonoplast in tobacco BY-2 cells under salt treatment was detected due to the downregulation of 17 AGPs’ transcripts, while AGP accumulation was observed in culture media, suggesting that AGPs might function as sodium carriers through vesicle trafficking [31]. However, the massive upregulation of AGPs was observed in tobacco BY-2 cells under NaCl stress [30]. 

Minor changes in leaf and root *Oe*P4H expression were observed for Lefkolia and Arvanitolia under saline conditions, while sensitive Koroneiki and Gaidourelia exhibited upregulation in leaves and no specific expression trend in roots. These data suggest that the leaves and roots activated distinct mechanisms to respond to salinity. In Arvanitolia, minor changes were detected in the expression levels of *Oe*AGPs and in the immunolocalization levels of JIM13-bound AGPs in roots under saline conditions. Moreover, the JIM13-bound fluorescent signal showed identical distribution patterns in control and salt-treated root sections after a 45-day treatment. These results indicate similar levels of AGP content in salt-treated and control plants, which might be associated with similar cortical cell size, shape and intercellular spaces. In Lefkolia, a stronger signal of JIM13-bound epitopes in root cortical cells under saline conditions was accompanied by an increase in *OeAGP10-like* expression in roots. However, the JIM13 epitopes showed a stronger signal in cortical cells but also a lower signal in rhizodermal cells under saline conditions. Therefore, the higher expression of *OeAGP10-like* in salt-treated roots might be attributed to the higher number of cortical cells that upregulated this mRNA compared to rhizodermal cells. In sensitive Koroneiki, the downregulation of both *OeAGP4-like* and *OeAGP10-like* after 45 days of salinity was accompanied by a lower signal of JIM13-bound epitopes. The fluorescent signal was weaker under saline conditions, mainly in the cortex, where the cells were arranged in a non-orderly way and their shape, size and intercellular spaces were variable. These data might indicate an association between AGP content and cortical cell structure and morphology, taking into consideration the known involvement of AGPs in cell size, enlargement and expansion [36,41]. 

In Arvanitolia, LM2 epitopes were induced in roots under saline conditions, while JIM13 epitopes were marginally downregulated in Gaidourelia roots under stress according to Western blot analysis. These results might indicate the induction of AGPs under saline conditions, particularly in tolerant cultivars, suggesting the association of AGPs with stress resilience. The immunolocalization of AGPs showed salinity-induced weak and strong signals in the cortices of Koroneiki and Lefkolia roots, respectively. In Arvanitolia under saline conditions, strong signals were detected in the root phloem and xylem. These data indicate that variation in AGP expression in the root cortex and stele is not consistent among cultivars. Due to this cell-type-dependent variation in AGP expression, Western-blot-based AGP content might not be considered comparable to immunodetection-based AGP levels in salinity and control roots.

Co-expression patterns of *OeAGP10-like* and *OeP4H1* in the roots of Gaidourelia and Koroneiki and in the leaves of Lefkolia were observed. This co-expression pattern in three cultivars and two plant organs might suggest *Oe*P4H1 specificity for the proline hydroxylation of *OeAGP10-like*. However, gene regulatory co-expression networks are required to identify co-expression patterns among olive P4Hs and substrate proteins such as AGPs, extensins and hormone peptides, which are known to be involved in abiotic stress responses [85]. 

Salinity causes the softening of the cell wall, and this decrease in stiffness was attributed to high Na+, which disrupts ionic interactions, such as the egg-box structures of pectins, in which Ca^2+^ ions and de-esterified homogalacturonans are major structural components [86]. These are considered load-bearing structures, and disruption by higher Na^+^ levels due to salinity stress results in the softening of the cell wall. The sensing of the disruption of these structures was attributed to the FERONIA (FER) receptor, which in turn initiates Ca^2+^ transient induction to restore defects and maintain cell wall integrity under salinity stress [86].

Uronic acids in terminal positions in AGPs are considered essential for Ca^2+^ binding, while AGPs bind Ca^2+^ more strongly than pectins [38]. There is also a stoichiometric relation between Ca^2+^ and the carboxyl groups of AGPs, as previously demonstrated [38], while the Arabinogalactan carbohydrate in AGPs was shown to bind and release Ca^2+^ at the cell surface [87]. Arabidopsis Arabinogalactan β-glucuronyltransferase mutants, which add glucuronic acids to AGPs, exhibited a decrease in the capacity to bind Ca^2+^, indicating a role as a putative Ca^2+^ capacitor [87]. Moreover, it was suggested that structural variations in the AGP structure, such as the O-methylation of monosaccharides, might affect the binding preference between Ca^2+^ and Na^+^ ions [38]. 

The role of AGPs might be considered important since Ca^2+^ binding by the uronic acids of AGPs possibly regulate their levels in the plasma membrane. In this context, the content and the subcellular localization of AGPs in olive roots might regulate Ca^2+^ fluxes under NaCl stress and, as a result, the response and tolerance to salinity. Salinity prevented the symplastic xylem from loading Ca^2+^ in the roots, resulting in the growth inhibition of leaves sensitive to salt [79]. The leaf acquires its final size due to cell division and cell elongation. Cell division, which regulates leaf initiation, was not affected by salt stress in sugar beet, but leaf expansion was a salt-sensitive process [88], depending on the Ca^2+^ status [89].

Alternatively, AGPs might function as carriers of Na^+^ and, after binding to the carboxyl groups of demethyl-esterified pectins, might be excluded from the vacuoles through vesicles as a mechanism to maintain Na^+^ contents at low levels within the cells [31]. The accumulation of AGPs in the xylem sap of Brassica has also been observed under salinity stress [33], which might suggest a role in Na^+^ ion translocation to the upper part of the plant.

Demethyl esterification of homogalacturonans (HGs) is important for pectin remodeling, which is needed for cell elongation [90]. Previous studies suggested that low pectin levels might result in the inhibition of root growth under saline conditions [91], while the abundance of methyl-esterified HGs slightly decreased in olive root cortical cell walls under saline conditions. 

In Koroneiki, decreases in the signal intensity of AGPs and methyl-esterified and demethyl-esterified HGs were observed under saline conditions, while in salt-stress-tolerant Lefkolia and Arvanitolia, the signal strength was similar to the control in salt-treated roots, with the exception of AGP upregulation in leaves. These results indicate an association between tolerance to salinity and the levels of AGPs and demethyl- and methyl-esterified HGs in roots. Recently, it was demonstrated that Rhamnogalacturonans-I (RGIs) were covalently linked to Arabinogalactan proteins (AGPs) in cultured Arabidopsis cell walls, indicating the strong interaction of pectins with AGPs [92]. Moreover, AGPs are considered to be part of tight complexes with cell wall polysaccharides through covalent and non-covalent linkages [93]. 

Moreover, an Arabidopsis FLA16 mutant, which showed a short-stem phenotype, exhibited lower levels of cellulase synthase gene expression, indicating the involvement of FLAs in cell wall biosynthesis and possibly the repair of cell wall defects due to salinity [94]. The Arabidopsis sos5/fla4 mutant showed swollen root tip cells under salt stress due to cell expansion defects [34]. Alterations in the cell wall and middle lamellae were also observed, while a synergistic role with ABA as an ABA signaling component was suggested [95]. In addition, Arabidopsis mutants of two AGP-specific galactosyltransferases (GALT2 and GALT4) and two cell-wall-associated leucine-rich repeat receptor-like kinase (FEI kinase) mutants showed phenotypes similar to sos5/fla4, indicating participation in the same signaling pathway [96].

In this context, the lower AGP content in the root cortex of Koroneiki might be involved in its higher sensitivity to NaCl stress. Moreover, specific gene expression patterns in P4Hs and AGPs were observed not only in roots but also in leaves, which might be related to the translocation of Na^+^ ions to the upper part of the tree in sensitive olive cultivars. Aerenchyma formation in the roots of Koroneiki under salinity treatment suggests that long-term salt stress might induce a hypoxia-related response in olive trees. The physiological significance of P4Hs and AGPs, as well as their involvement in signaling pathways under saline conditions, needs to be further investigated in olive trees.

Further research is required on the role of Ca^2+^ levels in the response of olive trees to salt stress, as well as studies on additional components of salinity-sensing machinery such as *sos5*/FLAs, AGP-specific galactosyltransferases and FERONIA in relation to the Na^+^ contents in roots and leaves and their putative translocation mechanisms.

## Figures and Tables

**Figure 1 cells-12-01466-f001:**
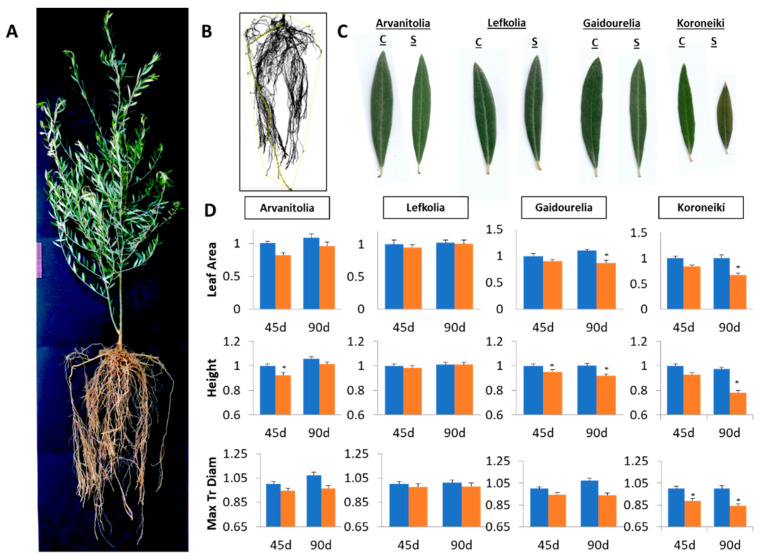
(**A**) Morphological characterization of leaves of Arvanitolia, Lefkolia, Koroneiki and Gaidourelia olive cultivars during salinity treatment time course and representative Koroneiki tree and root structure. (**B**) Representative structure of a 2-year-old Koroneiki tree grown in a 90 cm long barrel for 4 months and structure of the rooting system. (**C**) Representative olive leaves of the four cultivars in control (C) and salinity (S) treatments. (**D**) Leaf area, length and maximum transversal diameter (width) were determined after 45 (45 d) and 90 days (90 d)of salt treatment (Blue columns refer to control plants while orange columns to salt treated plants). Asterisks indicate statistically significant differences (*p*-value ≤ 0.01).

**Figure 2 cells-12-01466-f002:**
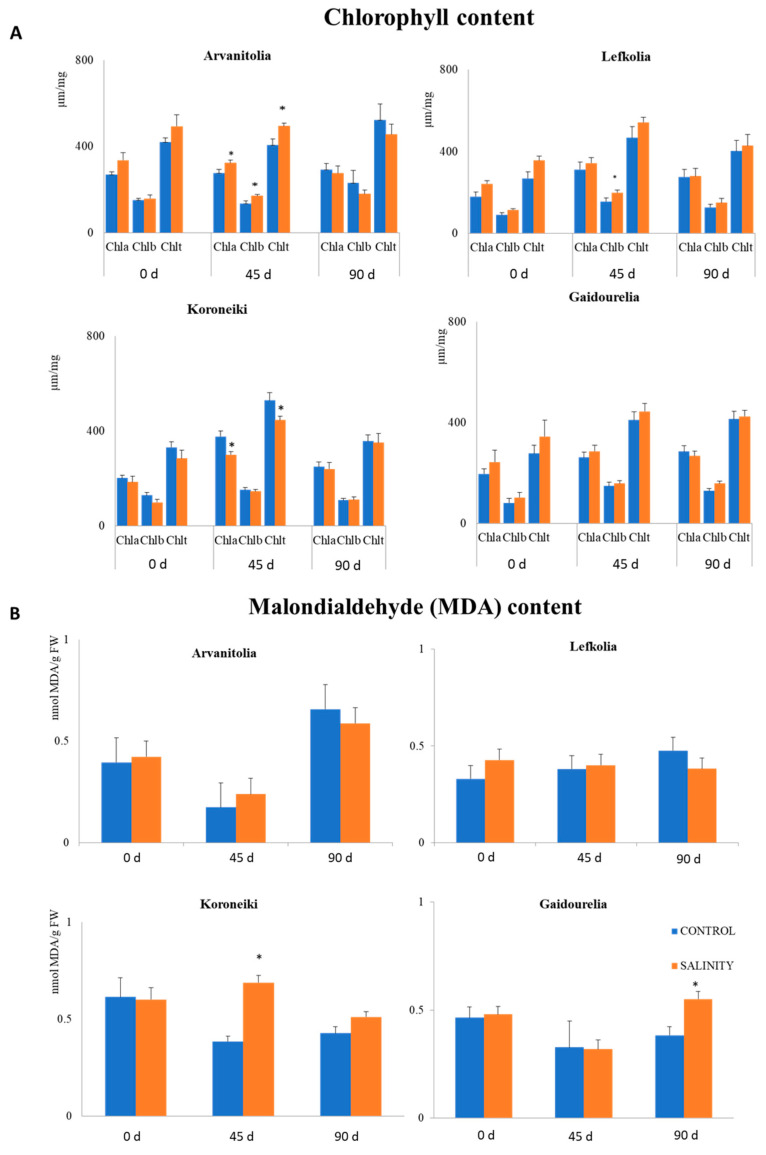
Chlorophyll and malondialdehyde contents in leaves of Arvanitolia, Lefkolia, Koroneiki and Gaidourelia during the salinity treatment time course. (**A**) Concentrations (μm/mg) of chlorophyll a (Chla), chlorophyll b (Chlb) and total chlorophyll (Chlt), expressed as μM per mg of fresh matter, were determined from leaves of four cultivar olive trees after 45 (45 d) and 90 days (90 d) of salt treatment. (**B**) Malondialdehyde (MDA) content was determined from leaves of four cultivar olive trees after 45 (45 d) and 90 days (90 d) of salt treatment. Asterisks indicate statistically significant differences (*p*-value ≤ 0.01). Zero days (0 d) refers to the untreated control.

**Figure 3 cells-12-01466-f003:**
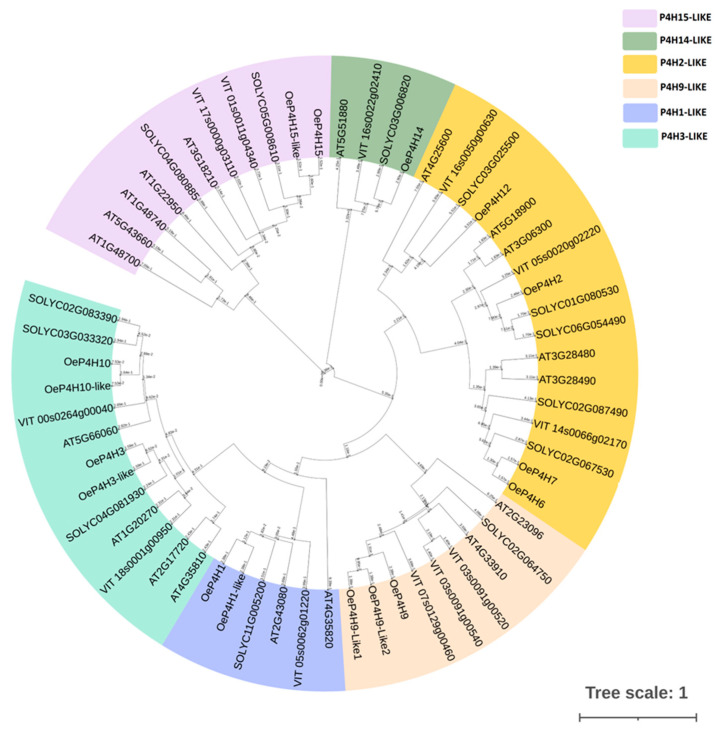
Phylogenetic tree of olive (*Oe*P4Hs), Arabidopsis (AT), tomato (SOLYC) and grapevine (VIT). The P4Hs were clustered into 6 groups according to the clustering of 16 putative Arabidopsis P4Hs. Branch length is also indicated. The accession numbers of olive P4Hs: *Oe*_P4H15 (LOC111396321), *Oe*_P4H15-like *Oe*_P4H12 (LOC111406102), *Oe*_P4H2 (LOC111382905), *Oe_*P4H6 (LOC111409222), *Oe*_P4H7 (LOC111397086), *Oe*_P4H9 (LOC111367228), *Oe*_P4H9-like1 (LOC111390203), *Oe*_P4H9-like2 (LOC111397020), *Oe*_P4H1 (LOC111397768), *Oe*_P4H1-like (LOC111384341), *Oe*_P4H3 (LOC111397368), *Oe*_P4H3-like (LOC111376712), *Oe*_P4H10 (LOC111408974) and *Oe*_P4H10-like (LOC111397201).

**Figure 4 cells-12-01466-f004:**
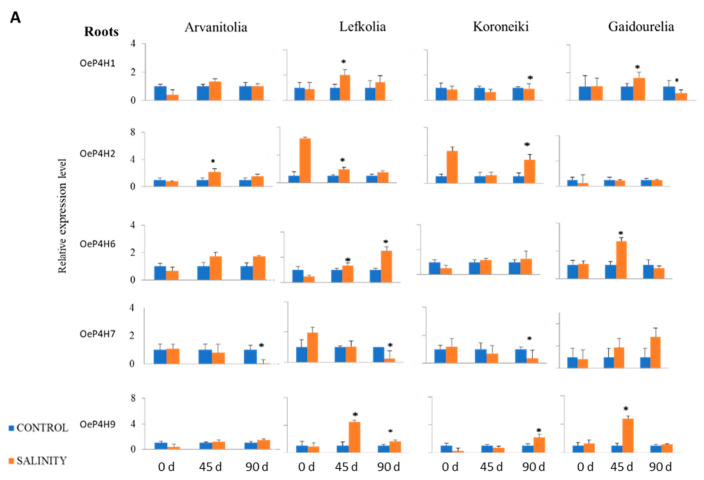
Relative expression of *OeP4H1*, *OeP4H2*, *OeP4H6*, *OeP4H7* and *OeP4H9* in roots and leaves of Arvanitolia, Lefkolia, Koroneiki and Gaidourelia olive cultivars during the salinity treatment time course. (**A**) Relative expression of *OeP4H1*, *OeP4H2*, *OeP4H6*, *OeP4H7* and *OeP4H9* in roots of four cultivars olive trees after 45 (45 d) and 90 days (90 d) of salt treatment. (**B**) Relative expression of *OeP4H1*, *OeP4H2*, *OeP4H6*, *OeP4H7* and *OeP4H9* in leaves of four cultivars olive trees after 45 and 90 days of salt treatment. The relative expression was calculated according to the comparative Ct method by using actin as an internal standard. Asterisks indicate statistically significant differences (*p*-value ≤ 0.01). Zero days (0 d) refers to the untreated control.

**Figure 5 cells-12-01466-f005:**
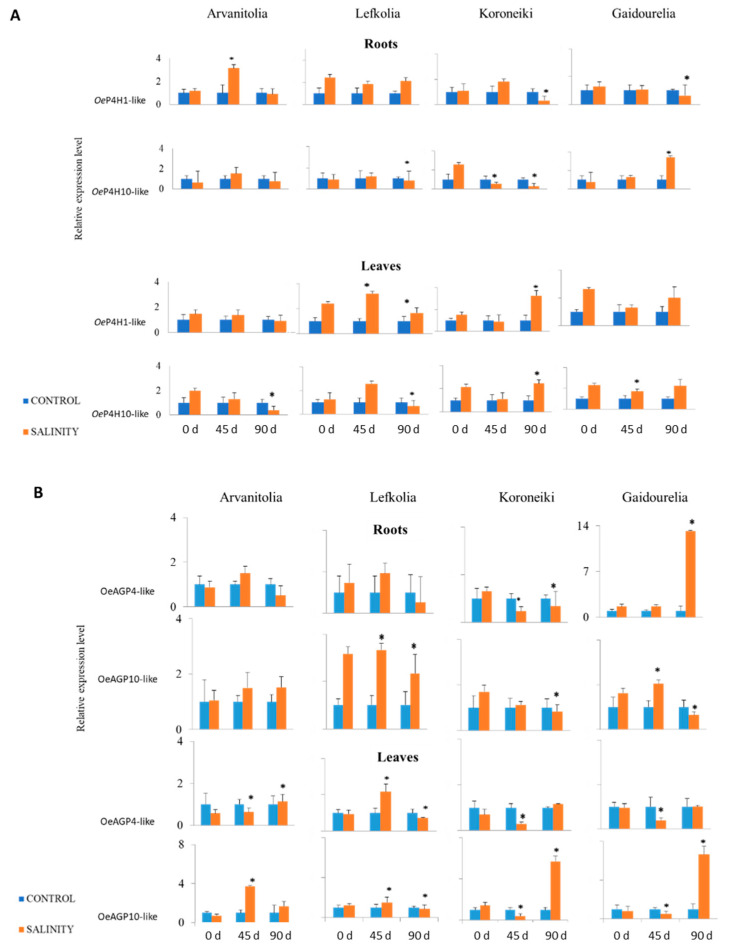
Relative expression of *Oe*P4H1-like and *Oe*P4H10-like in roots and leaves of Arvanitolia, Lefkolia, Koroneiki and Gaidourelia olive cultivars during the salinity treatment time course. (**A**) Relative expression of *Oe*P4H1-like and *Oe*P4H10-like in roots and leaves of four cultivars of olive trees after 45 (45) and 90 days (90) of salt treatment. (**B**) Relative expression of *Oe*AGP4-like and *Oe*AGP10-like in roots and leaves of four cultivars of olive trees after 45 (45 d) and 90 days (90 d) of salt treatment. The relative expression was calculated according to the comparative Ct method by using actin as an internal standard. Asterisks indicate statistically significant differences (*p*-value ≤ 0.01). Zero days (0 d) refers to the untreated control.

**Figure 6 cells-12-01466-f006:**
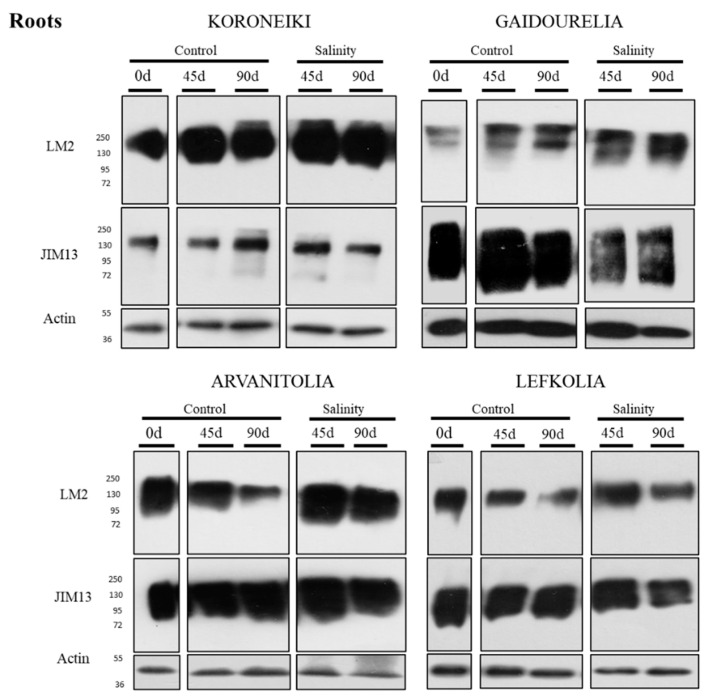
Western blot analysis of LM2- and JIM13-bound AGPs in roots of Arvanitolia, Lefkolia, Koroneiki and Gaidourelia olive cultivars during the salinity treatment time course. Total proteins were extracted from roots of four cultivars of olive trees after 45 (45 d) and 90 days (90 d) of control (untreated) and salinity treatments, and proteins (15 μg) from each sample were separated by SDS-PAGE prior to immunoblot analysis using LM2 and JIM13 antibodies and the actin antibody for the loading control. Zero days (0 d) refers to the untreated control. The molecular masses (kDa) are indicated on the left.

**Figure 7 cells-12-01466-f007:**
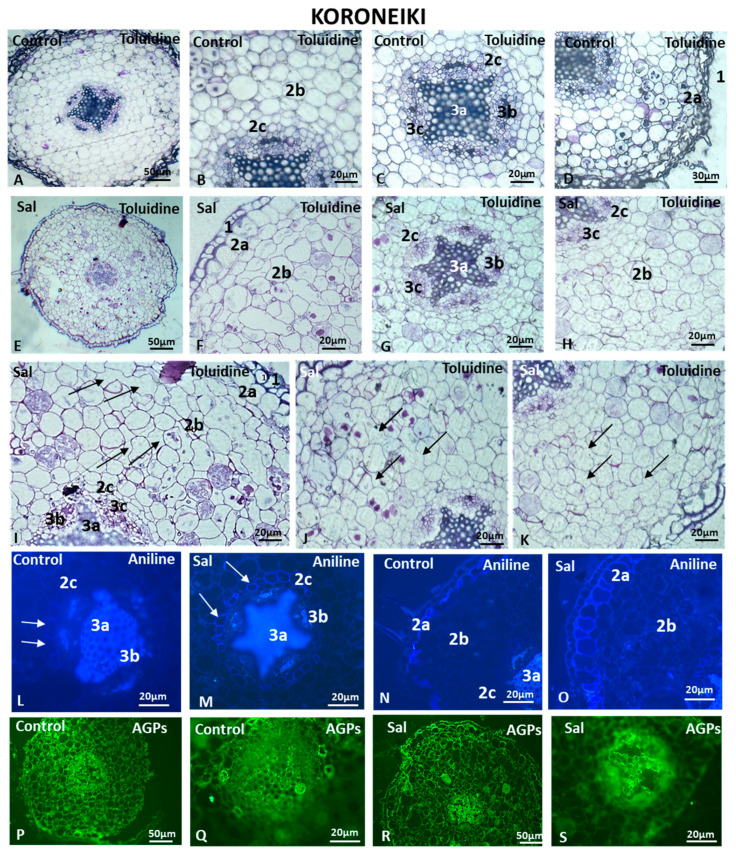
Structure, aniline and AGP content in Koroneiki roots untreated (Control) and under saline conditions (Sal). Samples stained with toluidine blue (**A**–**K**), aniline blue (**L**–**O**) and JIM13 antibody for immunolocalization (**P**–**S**). 1: Rhizodermis; 2a,b,c: cortex (2a: exodermis; 2b: parenchyma cortical cells; 2c: endodermis); 3a,b,c: vascular cylinder (3a: xylem; 3b: phloem; 3c: pericycle). Arrows in (**I**–**K**) indicate aerenchyma formation. Arrows in (**L**,**M**) indicate the endodermis. Three independent experiments were performed. Pictures are representative of the observations.

**Figure 8 cells-12-01466-f008:**
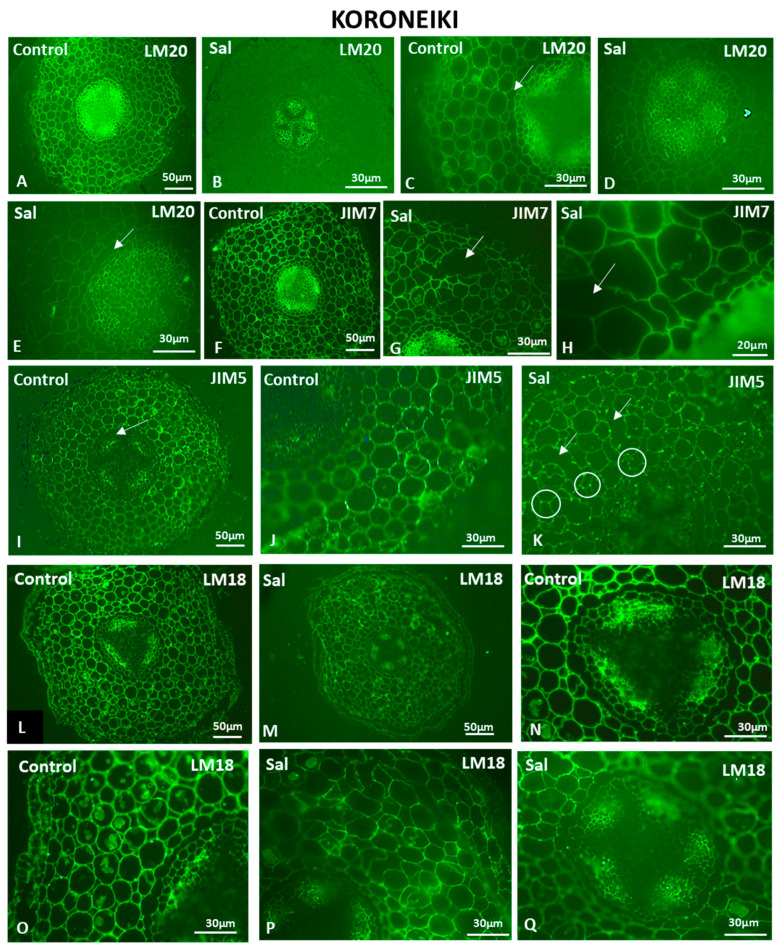
HG content in Koroneiki roots untreated (Control) and under saline conditions (Sal). Fully methyl-esterified (LM20) (**A**–**E**) and demethylesterified with a high degree of methylesterification (JIM7) (**F**–**G**) HG localization. Partially demethyl-esterified (JIM5: (**I**–**K**) and LM18: (**L**–**Q**)) HG localization. Arrows in (**C**,**E**,**I**): anticlinal cell walls of the endodermis. Arrows in (**G**,**H**): large intercellular space formation shown. Circles in (**K**): junction sites of adjacent cells displaying JIM5 fluorescent signal. Arrows in (**K**): Large intercellular spaces. Three independent experiments were performed. Pictures are representatives of the observations made.

**Figure 9 cells-12-01466-f009:**
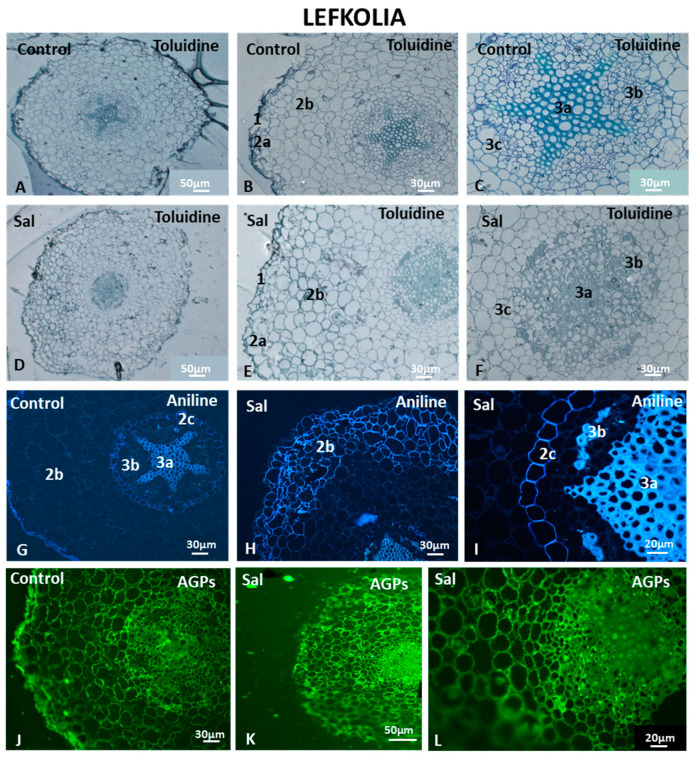
Structure, aniline and AGP content in Lefkolia roots untreated (Control) and under saline conditions (Sal). Samples stained with toluidine blue (**A**–**F**), aniline blue (**G**–**I**) and JIM13 antibody for immunolocalization (**J**–**L**). 1: Rhizodermis; 2a,b,c: cortex (2a: exodermis; 2b: parenchyma cortical cells; 2c: endodermis); 3a,b,c: vascular cylinder (3a: xylem; 3b: phloem; 3c: pericycle). Three independent experiments were performed. Pictures are representative of the observations.

**Figure 10 cells-12-01466-f010:**
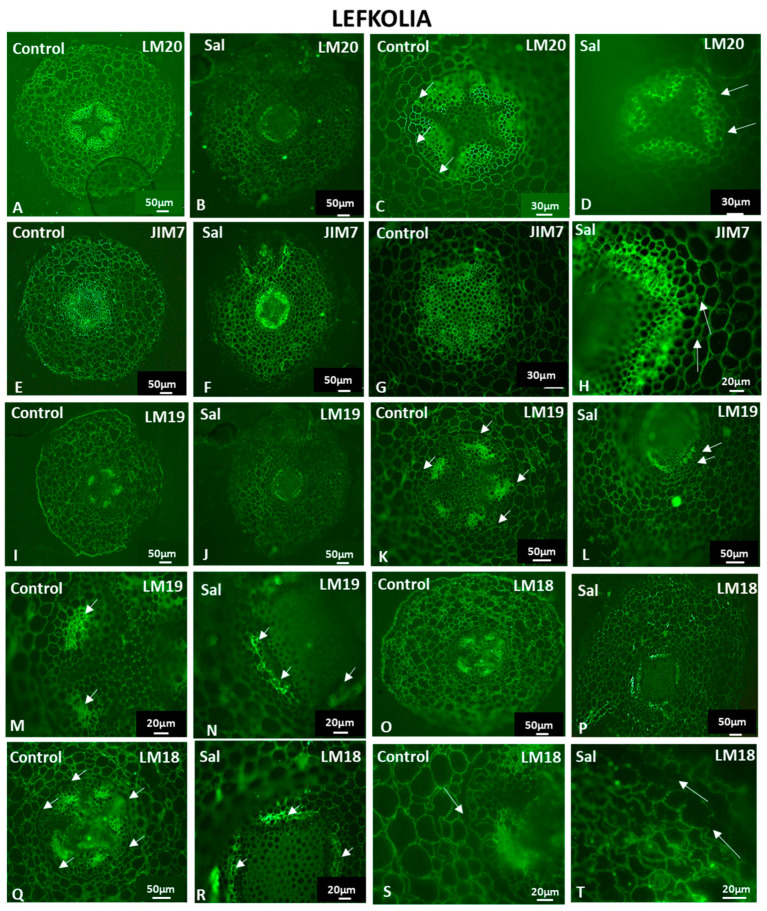
HG content in Lefkolia roots untreated (Control) and under saline conditions (Sal). Fully methyl-esterified (LM20) (**A**–**D**) and demethylesterified with a high degree of methylesterification (JIM7) (**E**–**H**) HG localization. Partially demethyl-esterified (LM19: **I**–**N** and LM18: **O**–**T**) HG localiza-tion. Arrows in (**C**,**D**,**H**,**K**,**L**,**Q**,**S**): anticlinal cell walls of the endodermis. Arrows in (**M**,**N**,**R**): phloem cells. Arrows in (**T**): anticlinal cell walls of the exodermis. Three independent experiments were performed. Pictures are representatives of the observations made.

**Figure 11 cells-12-01466-f011:**
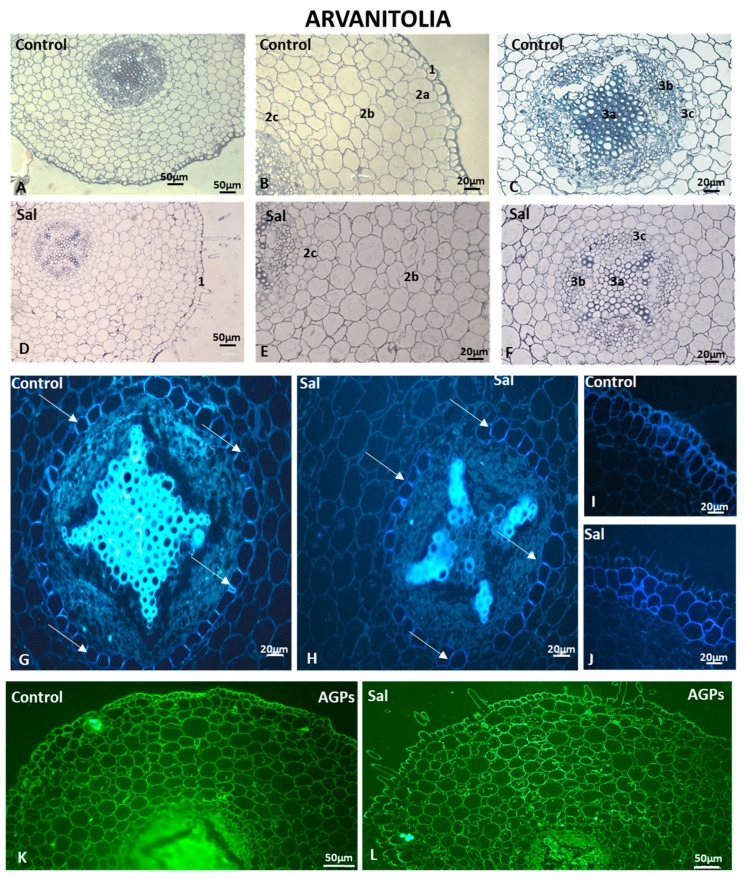
Structure, aniline and AGP content in Arvanitolia roots under saline conditions. Samples stained with toluidine blue (**A**–**F**), aniline blue (**G**–**J**) and JIM13 antibody for immunolocalization (**K**,**L**). 1: Rhizodermis; 2a,b,c: cortex (2a: exodermis; 2b: parenchyma cortical cells; 2c: endodermis); 3a,b,c: vascular cylinder (3a: xylem; 3b: phloem; 3c: pericycle). Arrows in (**G**,**H**) indicate the endodermis. Three independent experiments were performed. Pictures are representative of the observations.

**Figure 12 cells-12-01466-f012:**
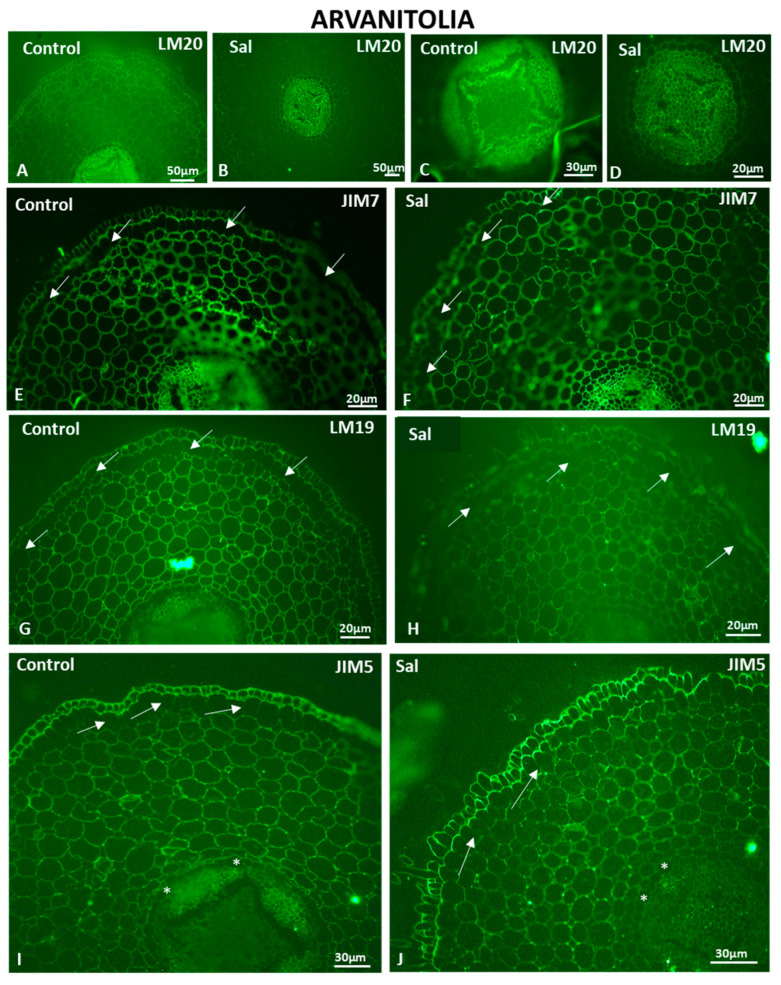
HG content in Arvanitolia roots untreated (Control) and under saline conditions (Sal). Fully methyl-esterified (LM20) (**A**–**D**) and demethylesterified with a high degree of methylesterification (JIM7) (**E**,**F**) HG localization. Partially demethyl-esterified (LM19: **G**,**H** and JIM5: **I**,**J**) HG localization. Arrows in (**E**–**J**): anticlinal cell walls of the exodermis. Asterisks in (**I**,**J**): endodermal cells. Three independent experiments were performed. Pictures are representatives of the observations made.

**Table 1 cells-12-01466-t001:** Leaf Chla/Chlb ratio in control and NaCl-treated olive trees.

Chl_a_/Ch_b_	Arvanitolia		Gaidourelia		Koroneiki		Lefkolia	
	45 Days	90 Days	45 Days	90 Days	45 Days	90 Days	45 Days	90 Days
Control	2.03	2.3	1.8	2.2	2.5	2.3	2.0	2.2
Salinity	1.9	2.5	1.2	1.1	1.6	1.1	2.2	2.7

## Data Availability

Not applicable.

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
