# Peer review of "Response of Prolyl 4 Hydroxylases, Arabinogalactan Proteins and Homogalacturonans in Four Olive Cultivars under Long-Term Salinity Stress in Relation to Physiological and Morphological Changes"

_cells, 2023, doi:10.3390/cells12111466_

Round 1

Reviewer 1 Report

1. Introduction, please provide more information or reviews to link between salinity and olive tree in the Mediterranean area, or how to link olive tree, Mediterranean basin, and salinity?

2. Introduction, please provide more specific information on effects of salinity on roots and leaves of olive tree (morphological and physical responses) from literature reviews.

3. Plant material, please indicate the differential tolerance of 4 olive cultivars with references.

4. Results, Figure1D, please improve quality of the picture, and indicate type of statistic use in a figure legend.   

5. Results, Figure 2A, what is the unit of chlorophyll content? (micro mole?).

Please indicate type of statistic use in a figure legend.  

6. Results, Figure 4, 5, Please indicate type of statistic use in a figure legend.  

7. Results, line 369, is it 90 days or 95 days??

8. Overall, the discussion section lacks the in-depth discussion/explanation. It needs the physiological, biochemical or molecular mechanisms of salinity stress response of the olive tree, and links to P4Hs, AGPs, leaf, root, chlorophyll, MDA, root structure, endodermis, and aerenchyma, comparing between salt tolerant and salt sensitive olive cultivars. Although, the authors mentioned that the physiological significance of P4Hs and AGPs needs to be further investigated under salinity, the novel knowledge needs to be presented in the present study.

Author Response

All the suggestions of the reviewer 1 were taken into consideration and were replied one by one. 

Reviewer 2 Report

General Comments for Authors

The research work is interesting and falling within the scope of the journal. The authors have done research with title “Response of prolyl 4 hydroxylases, Arabinogalactan proteins and pectin’s content under long salinity stress in four olive cultivars”. There is well consistency between the title of the article and the results, and their interpretation. However, improvements like sentence structure and short sentences would make the manuscript more effective. The article dataset certainly contain constructive information for scientific community.

The following points may be addressed by the Authors to enhance the worth of the paper. After the addition of these suggestions, I will recommend this manuscript for further process. I will never recommend to publish this manuscript in the current write-up.

Abstract

The abstract part looks good. I will suggest to describe main treatment and its results at the end of abstract which would be suggestable to farmers to get higher yield of olive under stress.  Author also needs to provide main results of his study findings.

Introduction

The write up of the introduction part is good. Author needs to provide data about percent reduction in yield of olive due to salinity. I will suggest to give data about olive uses, composition, annual production in world and top olive producing countries in introduction part.

Materials and Methods

The materials and methods are well presented. However, it need thorough improvement in sentence structure.

Results and discussion

The description of results and discussion part is also well. However, author needs to justify his study results in proper way by giving proper reasons and recent studies references. Author has used too much old references and citations.

References

There are many references cited which are too old studies. Author needs to replace them with latest studies.

Specific Comments

Line 22-25- Replace the sentence “and intercellular spaces under salinity were similar to control in Arvanitolia while in Koroneiki weak AGPs signal was correlated to irregular cells and intercellular spaces which was accompanied by aerenchyma formation after 45 days of NaCl-treatment.” by “and intercellular spaces under salinity were similar to control in Arvanitolia, while in Koroneiki weak AGPs signal was correlated to irregular cells and intercellular spaces which was accompanied by aerenchyma formation after 45 days of NaCl-treatment.”

Line 774-775- Too old reference. Author should replace it with recent studies.

Line 806, 811- The given reference is old.

Page 825-826- Replace the reference with latest study.

Page 848, 854, 856 and 858- All the references which has been cited by the author are too old. I will suggest to replace it with some latest study references.

Author Response

All the suggestions of reviewer 2 were taken into consideration and were replied one by one. 

Round 2

Reviewer 1 Report

Nice responses from the authors 

Author Response

Cells Revision by Jamet 21/11/2022

Dear Authors, I have carefully read your manuscript. I am sorry to say that, despite improvements after the reviewing process, there are still a lot of major flaws which need to be corrected before the manuscript can be considered for publication. For the moment, it will be put under the “reject and encourage re-submission” status. I have put notes on the manuscript and make a list of comments to help you in this task. Please carefully take them into account before re-submission and consider re-writing extensively the introduction and the discussion sections, while clarifying the results section.

Regards. E Jamet

The comments in the cells1953958 decision v1 pdf were answered one by one. In addition, all the corrections were incorporated in the revised version of the manuscript.

The English language needs to be improved.

We believe that the English language was improved.

Title

The title of the article is focused on cell-wall related components, but its content includes other information like morphological characterization of leaves and roots, physiological parameters like chlorophyll and malondialdehyde content measurements. This is confusing and the whole story should be re-organized to clarify the message, starting with the introduction.

The title was revised to represent the entire research output of the manuscript.

 Introduction

The introduction has to be shortened and better organized. Similar ideas should be put together and its content should be focused on the topic of the manuscript which remains to be clearly defined. The introduction should also include a short description of the cell walls as well as the role of homogalacturonans. The choice of the four studied cultivars should be more clearly explained. Finally, the purpose of the study has to be clearly stated. For me, the link between AGPs and HGs is not evident. Altogether, this manuscript collects a lot of data but the overall meaning is missing.

The Introduction was revised and shortened. The reasoning behind the selection of the four cultivars was clearly stated and a description of the homogalacturonans function and structure was presented. Recently, it was demonstrated that Ramnogalacturonans-I (RGIs) were covalently linked to Arabinogalactan proteins (AGPs) indicating strong interaction of pectins with AGPs (Li et al., (2023) Most of the rhamnogalacturonan-I from cultured Arabidopsis cell walls is covalently linked to arabinogalactan-protein, Carbohydrate Polymers 301 (2023) 120340)

  The objective of this work was clearly stated at the end of the introduction.

Material and Methods

The city and the country of the purchasers should be indicated at their first occurrence. The description of the antibodies could be replaced by a table provided in supplementary data. The description of the bioinformatics programs using to build up the phylogenic tree is missing as well as the reference of the Olive genome database. I could get access to the protein sequences at NCBI, but some of them cannot be found at Phytozome. It is also necessary to define the difference between P4H and P4H-like proteins. The methods used to identify the AGPs genes are missing as well as the accession numbers of the genes.

The city and the country of the purchasers were included in the manuscript. A table was created to include all the antibodies used in the manuscript. The description of the bioinformatics programs which were used to built the phylogenic tree was included in the Materials & Methods section.

The pipeline used to identify the olive AGPs was described in detail in the Materials & Methods section as well as the pertinent reference. The accessions numbers of the two olive AGPs were included in the Supplementary Table.

Results

Considering that Gaidourelia is a salt sensitive cultivar, we thought to concentrate on the two tolerant cultivars in order to characterize their molecular response to salinity at the anatomical and immunolocalization level.  The characterization of Koroneiki only was considered adequate for the comparison between tolerant and sensitive cultivars.

The percentages of identity among the olive P4Hs and P4Hs-like were included in the results section.

In the whole manuscript, levels of accumulation of transcripts and of proteins should be clearly distinguished. The gene names should be in italics everywhere.

The corrections were included in the manuscript according to the editor’s suggestions.

The length of roots were measured (§ 2.1). But how was it precisely done, considering the fact that the plants exhibit hairy roots? Wouldn’t the measurement of their mass be more relevant?

The determination of the roots length was indicative in order to suggest the length of the root at the time the experiment was performed. The length was determined by taking into consideration only the major roots and not hairy roots considering that this is a tree of two years old. 

I do not understand why there is a detailed analysis of the P4H family and not of the AGP family.

The expression of seven different classical AGPs cDNAs were determined in response to salinity stress, but expression was detected only in two of them. Therefore, we decided not to proceed with further analysis of olive AGPs, although we have identified most of the olive AGPs by using the pipeline described in the Materials & Methods.   

In supplementary data, there are tables describing the structure of P4Hs. In some cases, it seems that signal peptides cannot be predicted. Since the olive genome has only been recently sequenced and annotated, could it be possible that the structural annotation is not correct? For example OEP4H1 has no homologue at Phytozome. Are there any RNAseq data available to check it? It would be worth checking the different databases providing olive genomic and RNAseq sequences to strengthen the data (Supplementary Tables 1 and 2). Besides, clarify what is a cytoplasmic tail and it role in transport to the Golgi (line 370).

The olive P4Hs were identified based on sequence homology and the presence of important for enzymatic activity aminoacids.

The following text was incorporated in the results section regarding the cytoplasmic tail: The Cytoplasmic tail is comprised of 5-20 amino acids (Supplementary Table 2) and is located in the N-terminus of the protein. The Cytoplasmic tail consists of positive charged amino acids similar to the RXR motif or dibasic signal found in mammalian and yeast Endoplasmic Reticulum (ER) and Golgi proteins. Similar cytoplasmic domains have been found to be responsible for the transfer of prolyl 4 hydroxylases to the Golgi apparatus in plants. Mutant GFP prolyl 4 hydroxylase proteins in which the basic amino acids of the cytoplasmic tail were substituted with non-charged hydrophilic amino acids in tobacco BY-2 cells were found to be located in the ER unlike the control ones which were observed in both the ER and Golgi apparatus.

The quantification of AGP epitopes on western blots is questionable (Figure 6). Indeed, the signals are saturated. The same experiment should be done with shorter times of exposure. The differences in the level of accumulation of LM2 epitopes vs JIM13 epitopes in the four cultivars could be discussed. Were the same extracts stained with the Yariv reagent to make sure that AGPs are present in the protein extracts?

Three biological replicates of the Western blots were performed, the main figure and additional two replicates. We have to stress out that the sampling of the roots was performed in a non-destructive way from two years old olive trees in 90 cm long containers. Therefore, slight variability among the western blots biological replicates can be justified considering that the plants were not growing in a growth chamber in petri dishes which ensure high level repeatability of the experiments. Moreover, 5 olive trees per cultivar for salinity and another 5 olive trees for control experiments were used. The protein extracts were not stained with yariv reagent and no shorter times of exposure were used for the western blots, however, the repeatability of the AGPs-bound epitopes content trends after three biological replicates ensures the validity of the results.

Overall, the resolution of the figures could be improved and Figures 7, 8 and 9 are too small. Please annotate the figures accordingly to the comments given in the text. As it is, it is very difficult to see the results. In addition, I am not sure that the results can be quantified. It would be necessary to make sure that all the observations have been done in the same conditions and that several biological replicates have been performed. The comments indicated in part 3.6.1 are probably also relevant for the next two paragraphs (3.6.2 and 3.6.3. Altogether, these parts of the results need to be clarified and better illustrated. Figure 3. The lettering is too small. It is not possible to read the values representing the posterior probabilities indicating that the tree is correct. Figure 5. The legend of Panel B is not correct.

The whole construction of Figures has changed. We added Figures 10, 11 and 12 to better describe our results. Now, Figures 7 and 8 refer to Koroneiki, Figures 9 and 10 to Lefkolia and Figures 11 and 12 to Arvanitolia. The resolution of Figures is improved.

We have annotated the figures accordingly to the comments of the editor.

Three biological replicates have been performed for every experiment. The authors do not suggest that any quantification has been made. The results are described only under the scope of the presence or absence of a specific epitope signal in the various tissues of the root.

Discussion

I am sorry if I did not read the discussion in detail. Overall, its length could be reduced and it should be better focused on the actual results provided by the manuscript and be better organized to group related ideas. Its first paragraph could give an overview of the results. It should not repeat the Results section (e.g. lines 678-683). There are some shortcuts like line 815. The results do not show any link between a given P4H and a particular AGP and one should consider that the results obtained at the transcriptional level do not allow drawing conclusions about the level of accumulation of the corresponding proteins. Lines 857-859: to which result shown in the manuscript this comment refers to? FLAs are mentioned (lines 796-802). Are the AGPs studied in this manuscript classical AGPs or FLAs?

The discussion was revised and we believe that it is now more concise and solid.

The AGPs characterized in this manuscript are classical AGPs and not FLAs.

References

Check all the references and homogenize the format: - Decapitalize the tiltes - Put Latin names in italics - Journal names should all be either abbreviated or not. - doi should be indicated everywhere or nowhere. Some mistakes (there might be others: - Ref 10: pages (432-438) - Ref 40: pages missing (3346-3369) (remove p. tpc.00027). - Ref 47: pages missing (1802) - Ref 55: to be checked - Ref 57: 2017, 13, 111 - Ref 81: 149, dev200363

The suggested corrections were incorporated in the manuscript.

Cells Revision by Jamet 21/11/2022

Dear Authors, I have carefully read your manuscript. I am sorry to say that, despite improvements after the reviewing process, there are still a lot of major flaws which need to be corrected before the manuscript can be considered for publication. For the moment, it will be put under the “reject and encourage re-submission” status. I have put notes on the manuscript and make a list of comments to help you in this task. Please carefully take them into account before re-submission and consider re-writing extensively the introduction and the discussion sections, while clarifying the results section.

Regards. E Jamet

The comments in the cells1953958 decision v1 pdf were answered one by one. In addition, all the corrections were incorporated in the revised version of the manuscript.

The English language needs to be improved.

We believe that the English language was improved.

Title

The title of the article is focused on cell-wall related components, but its content includes other information like morphological characterization of leaves and roots, physiological parameters like chlorophyll and malondialdehyde content measurements. This is confusing and the whole story should be re-organized to clarify the message, starting with the introduction.

The title was revised to represent the entire research output of the manuscript.

 Introduction

The introduction has to be shortened and better organized. Similar ideas should be put together and its content should be focused on the topic of the manuscript which remains to be clearly defined. The introduction should also include a short description of the cell walls as well as the role of homogalacturonans. The choice of the four studied cultivars should be more clearly explained. Finally, the purpose of the study has to be clearly stated. For me, the link between AGPs and HGs is not evident. Altogether, this manuscript collects a lot of data but the overall meaning is missing.

The Introduction was revised and shortened. The reasoning behind the selection of the four cultivars was clearly stated and a description of the homogalacturonans function and structure was presented. Recently, it was demonstrated that Ramnogalacturonans-I (RGIs) were covalently linked to Arabinogalactan proteins (AGPs) indicating strong interaction of pectins with AGPs (Li et al., (2023) Most of the rhamnogalacturonan-I from cultured Arabidopsis cell walls is covalently linked to arabinogalactan-protein, Carbohydrate Polymers 301 (2023) 120340)

  The objective of this work was clearly stated at the end of the introduction.

Material and Methods

The city and the country of the purchasers should be indicated at their first occurrence. The description of the antibodies could be replaced by a table provided in supplementary data. The description of the bioinformatics programs using to build up the phylogenic tree is missing as well as the reference of the Olive genome database. I could get access to the protein sequences at NCBI, but some of them cannot be found at Phytozome. It is also necessary to define the difference between P4H and P4H-like proteins. The methods used to identify the AGPs genes are missing as well as the accession numbers of the genes.

The city and the country of the purchasers were included in the manuscript. A table was created to include all the antibodies used in the manuscript. The description of the bioinformatics programs which were used to built the phylogenic tree was included in the Materials & Methods section.

The pipeline used to identify the olive AGPs was described in detail in the Materials & Methods section as well as the pertinent reference. The accessions numbers of the two olive AGPs were included in the Supplementary Table.

Results

Considering that Gaidourelia is a salt sensitive cultivar, we thought to concentrate on the two tolerant cultivars in order to characterize their molecular response to salinity at the anatomical and immunolocalization level.  The characterization of Koroneiki only was considered adequate for the comparison between tolerant and sensitive cultivars.

The percentages of identity among the olive P4Hs and P4Hs-like were included in the results section.

In the whole manuscript, levels of accumulation of transcripts and of proteins should be clearly distinguished. The gene names should be in italics everywhere.

The corrections were included in the manuscript according to the editor’s suggestions.

The length of roots were measured (§ 2.1). But how was it precisely done, considering the fact that the plants exhibit hairy roots? Wouldn’t the measurement of their mass be more relevant?

The determination of the roots length was indicative in order to suggest the length of the root at the time the experiment was performed. The length was determined by taking into consideration only the major roots and not hairy roots considering that this is a tree of two years old. 

I do not understand why there is a detailed analysis of the P4H family and not of the AGP family.

The expression of seven different classical AGPs cDNAs were determined in response to salinity stress, but expression was detected only in two of them. Therefore, we decided not to proceed with further analysis of olive AGPs, although we have identified most of the olive AGPs by using the pipeline described in the Materials & Methods.   

In supplementary data, there are tables describing the structure of P4Hs. In some cases, it seems that signal peptides cannot be predicted. Since the olive genome has only been recently sequenced and annotated, could it be possible that the structural annotation is not correct? For example OEP4H1 has no homologue at Phytozome. Are there any RNAseq data available to check it? It would be worth checking the different databases providing olive genomic and RNAseq sequences to strengthen the data (Supplementary Tables 1 and 2). Besides, clarify what is a cytoplasmic tail and it role in transport to the Golgi (line 370).

The olive P4Hs were identified based on sequence homology and the presence of important for enzymatic activity aminoacids.

The following text was incorporated in the results section regarding the cytoplasmic tail: The Cytoplasmic tail is comprised of 5-20 amino acids (Supplementary Table 2) and is located in the N-terminus of the protein. The Cytoplasmic tail consists of positive charged amino acids similar to the RXR motif or dibasic signal found in mammalian and yeast Endoplasmic Reticulum (ER) and Golgi proteins. Similar cytoplasmic domains have been found to be responsible for the transfer of prolyl 4 hydroxylases to the Golgi apparatus in plants. Mutant GFP prolyl 4 hydroxylase proteins in which the basic amino acids of the cytoplasmic tail were substituted with non-charged hydrophilic amino acids in tobacco BY-2 cells were found to be located in the ER unlike the control ones which were observed in both the ER and Golgi apparatus.

The quantification of AGP epitopes on western blots is questionable (Figure 6). Indeed, the signals are saturated. The same experiment should be done with shorter times of exposure. The differences in the level of accumulation of LM2 epitopes vs JIM13 epitopes in the four cultivars could be discussed. Were the same extracts stained with the Yariv reagent to make sure that AGPs are present in the protein extracts?

Three biological replicates of the Western blots were performed, the main figure and additional two replicates. We have to stress out that the sampling of the roots was performed in a non-destructive way from two years old olive trees in 90 cm long containers. Therefore, slight variability among the western blots biological replicates can be justified considering that the plants were not growing in a growth chamber in petri dishes which ensure high level repeatability of the experiments. Moreover, 5 olive trees per cultivar for salinity and another 5 olive trees for control experiments were used. The protein extracts were not stained with yariv reagent and no shorter times of exposure were used for the western blots, however, the repeatability of the AGPs-bound epitopes content trends after three biological replicates ensures the validity of the results.

Overall, the resolution of the figures could be improved and Figures 7, 8 and 9 are too small. Please annotate the figures accordingly to the comments given in the text. As it is, it is very difficult to see the results. In addition, I am not sure that the results can be quantified. It would be necessary to make sure that all the observations have been done in the same conditions and that several biological replicates have been performed. The comments indicated in part 3.6.1 are probably also relevant for the next two paragraphs (3.6.2 and 3.6.3. Altogether, these parts of the results need to be clarified and better illustrated. Figure 3. The lettering is too small. It is not possible to read the values representing the posterior probabilities indicating that the tree is correct. Figure 5. The legend of Panel B is not correct.

The whole construction of Figures has changed. We added Figures 10, 11 and 12 to better describe our results. Now, Figures 7 and 8 refer to Koroneiki, Figures 9 and 10 to Lefkolia and Figures 11 and 12 to Arvanitolia. The resolution of Figures is improved.

We have annotated the figures accordingly to the comments of the editor.

Three biological replicates have been performed for every experiment. The authors do not suggest that any quantification has been made. The results are described only under the scope of the presence or absence of a specific epitope signal in the various tissues of the root.

Discussion

I am sorry if I did not read the discussion in detail. Overall, its length could be reduced and it should be better focused on the actual results provided by the manuscript and be better organized to group related ideas. Its first paragraph could give an overview of the results. It should not repeat the Results section (e.g. lines 678-683). There are some shortcuts like line 815. The results do not show any link between a given P4H and a particular AGP and one should consider that the results obtained at the transcriptional level do not allow drawing conclusions about the level of accumulation of the corresponding proteins. Lines 857-859: to which result shown in the manuscript this comment refers to? FLAs are mentioned (lines 796-802). Are the AGPs studied in this manuscript classical AGPs or FLAs?

The discussion was revised and we believe that it is now more concise and solid.

The AGPs characterized in this manuscript are classical AGPs and not FLAs.

References

Check all the references and homogenize the format: - Decapitalize the tiltes - Put Latin names in italics - Journal names should all be either abbreviated or not. - doi should be indicated everywhere or nowhere. Some mistakes (there might be others: - Ref 10: pages (432-438) - Ref 40: pages missing (3346-3369) (remove p. tpc.00027). - Ref 47: pages missing (1802) - Ref 55: to be checked - Ref 57: 2017, 13, 111 - Ref 81: 149, dev200363

The suggested corrections were incorporated in the manuscript.

Cells Revision by Jamet 21/11/2022

Dear Authors, I have carefully read your manuscript. I am sorry to say that, despite improvements after the reviewing process, there are still a lot of major flaws which need to be corrected before the manuscript can be considered for publication. For the moment, it will be put under the “reject and encourage re-submission” status. I have put notes on the manuscript and make a list of comments to help you in this task. Please carefully take them into account before re-submission and consider re-writing extensively the introduction and the discussion sections, while clarifying the results section.

Regards. E Jamet

The comments in the cells1953958 decision v1 pdf were answered one by one. In addition, all the corrections were incorporated in the revised version of the manuscript.

The English language needs to be improved.

We believe that the English language was improved.

Title

The title of the article is focused on cell-wall related components, but its content includes other information like morphological characterization of leaves and roots, physiological parameters like chlorophyll and malondialdehyde content measurements. This is confusing and the whole story should be re-organized to clarify the message, starting with the introduction.

The title was revised to represent the entire research output of the manuscript.

 Introduction

The introduction has to be shortened and better organized. Similar ideas should be put together and its content should be focused on the topic of the manuscript which remains to be clearly defined. The introduction should also include a short description of the cell walls as well as the role of homogalacturonans. The choice of the four studied cultivars should be more clearly explained. Finally, the purpose of the study has to be clearly stated. For me, the link between AGPs and HGs is not evident. Altogether, this manuscript collects a lot of data but the overall meaning is missing.

The Introduction was revised and shortened. The reasoning behind the selection of the four cultivars was clearly stated and a description of the homogalacturonans function and structure was presented. Recently, it was demonstrated that Ramnogalacturonans-I (RGIs) were covalently linked to Arabinogalactan proteins (AGPs) indicating strong interaction of pectins with AGPs (Li et al., (2023) Most of the rhamnogalacturonan-I from cultured Arabidopsis cell walls is covalently linked to arabinogalactan-protein, Carbohydrate Polymers 301 (2023) 120340)

  The objective of this work was clearly stated at the end of the introduction.

Material and Methods

The city and the country of the purchasers should be indicated at their first occurrence. The description of the antibodies could be replaced by a table provided in supplementary data. The description of the bioinformatics programs using to build up the phylogenic tree is missing as well as the reference of the Olive genome database. I could get access to the protein sequences at NCBI, but some of them cannot be found at Phytozome. It is also necessary to define the difference between P4H and P4H-like proteins. The methods used to identify the AGPs genes are missing as well as the accession numbers of the genes.

The city and the country of the purchasers were included in the manuscript. A table was created to include all the antibodies used in the manuscript. The description of the bioinformatics programs which were used to built the phylogenic tree was included in the Materials & Methods section.

The pipeline used to identify the olive AGPs was described in detail in the Materials & Methods section as well as the pertinent reference. The accessions numbers of the two olive AGPs were included in the Supplementary Table.

Results

Considering that Gaidourelia is a salt sensitive cultivar, we thought to concentrate on the two tolerant cultivars in order to characterize their molecular response to salinity at the anatomical and immunolocalization level.  The characterization of Koroneiki only was considered adequate for the comparison between tolerant and sensitive cultivars.

The percentages of identity among the olive P4Hs and P4Hs-like were included in the results section.

In the whole manuscript, levels of accumulation of transcripts and of proteins should be clearly distinguished. The gene names should be in italics everywhere.

The corrections were included in the manuscript according to the editor’s suggestions.

The length of roots were measured (§ 2.1). But how was it precisely done, considering the fact that the plants exhibit hairy roots? Wouldn’t the measurement of their mass be more relevant?

The determination of the roots length was indicative in order to suggest the length of the root at the time the experiment was performed. The length was determined by taking into consideration only the major roots and not hairy roots considering that this is a tree of two years old. 

I do not understand why there is a detailed analysis of the P4H family and not of the AGP family.

The expression of seven different classical AGPs cDNAs were determined in response to salinity stress, but expression was detected only in two of them. Therefore, we decided not to proceed with further analysis of olive AGPs, although we have identified most of the olive AGPs by using the pipeline described in the Materials & Methods.   

In supplementary data, there are tables describing the structure of P4Hs. In some cases, it seems that signal peptides cannot be predicted. Since the olive genome has only been recently sequenced and annotated, could it be possible that the structural annotation is not correct? For example OEP4H1 has no homologue at Phytozome. Are there any RNAseq data available to check it? It would be worth checking the different databases providing olive genomic and RNAseq sequences to strengthen the data (Supplementary Tables 1 and 2). Besides, clarify what is a cytoplasmic tail and it role in transport to the Golgi (line 370).

The olive P4Hs were identified based on sequence homology and the presence of important for enzymatic activity aminoacids.

The following text was incorporated in the results section regarding the cytoplasmic tail: The Cytoplasmic tail is comprised of 5-20 amino acids (Supplementary Table 2) and is located in the N-terminus of the protein. The Cytoplasmic tail consists of positive charged amino acids similar to the RXR motif or dibasic signal found in mammalian and yeast Endoplasmic Reticulum (ER) and Golgi proteins. Similar cytoplasmic domains have been found to be responsible for the transfer of prolyl 4 hydroxylases to the Golgi apparatus in plants. Mutant GFP prolyl 4 hydroxylase proteins in which the basic amino acids of the cytoplasmic tail were substituted with non-charged hydrophilic amino acids in tobacco BY-2 cells were found to be located in the ER unlike the control ones which were observed in both the ER and Golgi apparatus.

The quantification of AGP epitopes on western blots is questionable (Figure 6). Indeed, the signals are saturated. The same experiment should be done with shorter times of exposure. The differences in the level of accumulation of LM2 epitopes vs JIM13 epitopes in the four cultivars could be discussed. Were the same extracts stained with the Yariv reagent to make sure that AGPs are present in the protein extracts?

Three biological replicates of the Western blots were performed, the main figure and additional two replicates. We have to stress out that the sampling of the roots was performed in a non-destructive way from two years old olive trees in 90 cm long containers. Therefore, slight variability among the western blots biological replicates can be justified considering that the plants were not growing in a growth chamber in petri dishes which ensure high level repeatability of the experiments. Moreover, 5 olive trees per cultivar for salinity and another 5 olive trees for control experiments were used. The protein extracts were not stained with yariv reagent and no shorter times of exposure were used for the western blots, however, the repeatability of the AGPs-bound epitopes content trends after three biological replicates ensures the validity of the results.

Overall, the resolution of the figures could be improved and Figures 7, 8 and 9 are too small. Please annotate the figures accordingly to the comments given in the text. As it is, it is very difficult to see the results. In addition, I am not sure that the results can be quantified. It would be necessary to make sure that all the observations have been done in the same conditions and that several biological replicates have been performed. The comments indicated in part 3.6.1 are probably also relevant for the next two paragraphs (3.6.2 and 3.6.3. Altogether, these parts of the results need to be clarified and better illustrated. Figure 3. The lettering is too small. It is not possible to read the values representing the posterior probabilities indicating that the tree is correct. Figure 5. The legend of Panel B is not correct.

The whole construction of Figures has changed. We added Figures 10, 11 and 12 to better describe our results. Now, Figures 7 and 8 refer to Koroneiki, Figures 9 and 10 to Lefkolia and Figures 11 and 12 to Arvanitolia. The resolution of Figures is improved.

We have annotated the figures accordingly to the comments of the editor.

Three biological replicates have been performed for every experiment. The authors do not suggest that any quantification has been made. The results are described only under the scope of the presence or absence of a specific epitope signal in the various tissues of the root.

Discussion

I am sorry if I did not read the discussion in detail. Overall, its length could be reduced and it should be better focused on the actual results provided by the manuscript and be better organized to group related ideas. Its first paragraph could give an overview of the results. It should not repeat the Results section (e.g. lines 678-683). There are some shortcuts like line 815. The results do not show any link between a given P4H and a particular AGP and one should consider that the results obtained at the transcriptional level do not allow drawing conclusions about the level of accumulation of the corresponding proteins. Lines 857-859: to which result shown in the manuscript this comment refers to? FLAs are mentioned (lines 796-802). Are the AGPs studied in this manuscript classical AGPs or FLAs?

The discussion was revised and we believe that it is now more concise and solid.

The AGPs characterized in this manuscript are classical AGPs and not FLAs.

References

Check all the references and homogenize the format: - Decapitalize the tiltes - Put Latin names in italics - Journal names should all be either abbreviated or not. - doi should be indicated everywhere or nowhere. Some mistakes (there might be others: - Ref 10: pages (432-438) - Ref 40: pages missing (3346-3369) (remove p. tpc.00027). - Ref 47: pages missing (1802) - Ref 55: to be checked - Ref 57: 2017, 13, 111 - Ref 81: 149, dev200363

The suggested corrections were incorporated in the manuscript.
